# PROSPECTIVE DECISION LEARNING FOR SAFE EXPLORATION IN MODEL-FREE REINFORCEMENT LEARNING

## ABSTRACT

Prospective thinking (PT) is the inherent ability of human beings, which guides the ahead-planning for decision making, becoming the key to efficient actions. However, current reinforcement learning methods lack PT in decision learning, leading to state traps caused by the lack of planning ahead, further reducing the data efficiency. This paper proposes a novel ProSpec RL method, which is the first to incorporate prospective decision learning to model-free RL for efficient and safe exploration. Specifically, to incorporate PT into model-free RL, we propose a flow-based reversible dynamics model, which predicts future n-stream trajectories based on the current state and policy. Meanwhile, to prevent the entrapment in state traps, we propose a prospective mechanism using model predictive control with value consistency constraint, enabling the learning to plan ahead then execute, to avoid "dead ends" caused by high-risk actions. Additionally, to improve data efficiency, we present a cyclical consistency constraint, which generates a large number of accurate and reversible virtual trajectories to further enhance state feature representations. Comprehensive evaluations of ProSpec on DMControl and Atari benchmarks demonstrate the significant accelerations in the model decision learning and the state-of-the-art performance in 4 of 6 DMControl and 7 of 26 Atari games. The code can be seen in the https://anonymous.4open.science/r/ProSpec-35B8/.

## 1 INTRODUCTION

Reinforcement learning (RL) has been proven to be indispensable for continuous decision-making problems (Mnih et al., 2015; Silver et al., 2017; Liu et al., 2024). The achievement of human-expert performance demonstrates the success of RL in critical scenarios such as autonomous driving and medical-assisted surgery (Vinyals et al., 2019; Kiran et al., 2021). Model-free methods are one dominant stream in RL, which optimize policy functions through trial-and-error, performing remarkable adaptability and delivering superiority in continuous control tasks (Hansen et al., 2022). However, the trial-and-error exploration in decision learning presents the model-free methods with severe challenges, such as data inefficiency and entrapment in state traps.

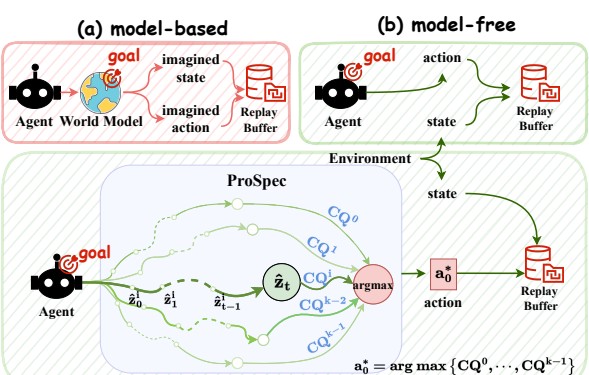

Figure 1: ProSpec vs. other methods. Model-based methods learn a world model to directly determine actions. Model-free methods, including ProSpec, aim to learn a policy. ProSpec simultaneously imagines K potential trajectories, evaluates their discounted returns $CQ_i$, and selects the initial action with the highest return, reducing risk and avoiding dangerous state traps from trial-and-error learning.

To address these challenges, current works proposed various effective methods. For instance, leveraging data augmentation techniques to increase the diversity of image appearances (Yarats et al., 2020; Laskin et al., 2020b), introducing auxiliary tasks to assist state representation learning (Schrittwieser et al., 2020; Schwarzer et al., 2021; Yu et al., 2021; Yue et al., 2023), using world models to assist

policy learning (Yuan et al., 2024; Zhu et al., 2024) to improve data efficiency, and exploring safer decision-making through contrasting unsafe states to build safe learning frameworks (Marchesini et al., 2022; Hu et al., 2023). These methods mainly focus on reducing mistakes and surviving traps in trial-and-error explorations by enhancing the characterization of data, features and policy optimization, achieving certain improvements, but struggle to avoid it by experience-based precautions due to some limitations in the simulation of human learning.

The inefficiency and state traps of trial-and-error processes rarely occur in human decision learning because humans have the ability of PT, which induces ahead planning before execution of each decision based on their experience. Compared with learning policies, ahead planning is considered more critical (Janner et al., 2019; Hansen et al., 2022). In the field of cognitive neuroscience and psychology, planning is a part of PT (Schacter et al., 2012), which aids in predicting and adapting to future changes, envisioning potential dangers, and setting achievable goals to motivate actions (Aspinwall, 2005). In real cases, PT is pivotal for continuous decision-making, which enables humans to envision future trajectories, makes more strategic decisions through planning, and chooses wisely when balancing short-term versus long-term benefits (Schacter et al., 2012). Evidently, the PT induced ahead planning is the key to avoiding trial-and-error learning for model-free RL methods but such mechanisms have been rarely proposed.

In this paper, we propose a novel ProSpec method, illustrated in Figure 1, which is the first to incorporate prospective decision learning to model-free RL for efficient and safe exploration. Specifically, to incorporate PT into model-free RL, we propose a flow-based reversible dynamics model (FDM), which predicts future n-stream trajectories based on the current state and policy, thereby expanding the model's foresight. Meanwhile, to prevent the entrapment in state traps, we present a prospective mechanism using a model predictive control (MPC) with a value-consistency constraint. With this mechanism, the ProSpec executes decisions after the ahead plan, avoiding high-risk actions of falling into "dead ends". Additionally, to enhance data efficiency, we present a cyclical consistency constraint that generates a large number of accurate and reversible virtual trajectories to further enhance state feature representations. The proposed ProSpec is comprehensively evaluated on two openly public benchmarks: DMControl and Atari. The results show that the ProSpec not only significantly accelerates the model decision learning with higher data efficiency and lower stuck actions, but also outperforms state-of-the-art methods in 4 of 6 DMControl and 7 of 26 Atari games.

The contributions of this paper are summarized as follows:

- We propose a novel ProSpec RL method, which is the first to incorporate PT into model-free RL, simulating the prediction of future n-stream trajectories based on the current state and policy through the FDM.
- We propose a prospective mechanism, whose ahead plans before execution effectively prevents the entrapment in state traps and avoids high-risk actions of falling into "dead ends" by an MPC with a value-consistency constraint.
- We present a cyclical consistency constraint for enhancing data efficiency by improving state feature representations from generated accurate and reversible virtual trajectories.

## 2 RELATED WORK

### 2.1 DATA EFFICIENCY IMPROVEMENT

Data efficiency has consistently been a major challenge hindering the advancement of RL. The early methods primarily focused on the trade-off between "exploration vs. exploitation" and "long-term vs. short-term reward" to improve data efficiency. For example, some more efficient strategies and enhanced policy update functions have been proposed, including $\epsilon$-greedy (Tokic, 2010), UCB (Slivkins et al., 2019), Temporal Difference Learning (Tesauro, 1991), Monte Carlo methods (Lazaric et al., 2007), and the Advantage Function (Baird, 1994).

With the advancement of deep learning, high-dimensional state and action spaces have progressively become the primary bottleneck hindering data efficiency. For example, SiMPLe (Kaiser et al., 2020) learns a world model from collected data and generates imagined trajectories to enhance data efficiency. DrQ (Yarats et al., 2021) and RAD (Laskin et al., 2020b) show that moderate image augmentations can significantly boost data efficiency in RL, even surpassing model-based methods.

Meanwhile, SPR (Hansen et al., 2022), PBL (Guo et al., 2020), and SLAC (Lee et al., 2020) have improved feature representation learning by incorporating dynamics models. In addition, recent research also shows that cyclical consistency effectively enhances the representation learning of dynamics models, thereby improving the data efficiency (Yu et al., 2021; Yue et al., 2023).

All these studies effectively improve the data efficiency of model-free methods, but overlook the modeling of human's PT. When confronted with data inefficiency, humans possess the ability of PT to plan ahead for dynamic adjustments before making decisions. This strategy minimizes unnecessary trial-and-error, thereby improving data efficiency. Building on this insight, the proposed method incorporates PT-guided ahead planning into model-free methods for improving data efficiency in decision-making learning.

## 2.2 State Trap Escapement

Due to the incomplete understanding of the environment, model-free methods are susceptible to state traps resulting from high-risk actions. For instance, in sparse reward settings, a positive reward may only be obtained at the last moment, which challenges the decision-learning of the model-free methods. One straightforward solution is to adjust the discount factor or optimize the reward function (Marvi & Kiumarsi, 2021; Yang et al., 2023) but struggle to balance performance and safety. The insufficiency of the decision penalty may fail to avoid risks, whereas the excess of the decision penalty will result in the cowardice of the agent's decision. Furthermore, to avoid high-risk trial-and-error decision learning, various strategies have been proposed to enhance decision safety, such as Lagrangian relaxation (Jayant & Bhatnagar, 2022), quadratic constraint optimization (Kim et al., 2024), and state-specific safety constraints (Zhang et al., 2023).

All these strategies mitigate the trouble of state traps but do not effectively alleviate the reliance on historical experience during policy updating of the model-free methods. In addition, from a human cognitive prospective, the absence of PT may also lead to state traps. Inspired by human cognition, we propose a prospective mechanism of "plan ahead, then execute" to avoid state traps.

## 3 Preliminaries: Reinforcement Learning

RL typically uses Markov decision processes (MDPs) to solve continuous decision problems, denoted as $< \mathcal{S}, \mathcal{A}, \mathcal{T}, \mathcal{R}, \gamma >$. Here $\mathcal{S}$ represents a finite set of states; $\mathcal{A}$ is the action space; $\mathcal{T}(s_t, a_t, s_{t+1}) = P(s_{t+1}|s_t, a_t)$ is the dynamic function, which defines the probability of transition from state $s_t$ to $s_{t+1}$ after taking action $a_t$; $\mathcal{R}(s_t, a_t)$ is the reward function, and $\gamma \in (0, 1]$ is the discount factor. The goal of RL is to enable the agent to learn how to maximize the expected cumulative discounted return $G_t = \sum_{\tau=0}^{\infty} \gamma^\tau \mathcal{R}(s_{t+\tau}, a_{t+\tau})$ by choosing actions at each time $t$. The agent's decision process is guided by the policy $\pi(a_t|s_t)$, which maps the current state to the action selection. The action-value function $Q_\pi(s_t, a_t) = \mathbb{E}_\pi[G_t|s_t, a_t]$ evaluates the expected return of taking action $a_t$ in state $s_t$ and following policy $\pi$. This paper focuses on value-based RL methods, specifically Q-learning, which approximates the optimal policy $\pi^*$ using the Bellman equation (Sutton & Barto, 2018). SAC is employed as the policy algorithm, as it is a widely used gradient-based method for continuous control that incorporates policy entropy as an additional reward to encourage exploration. The overall training objective of SAC is as follows:

$$\mathcal{J}_\theta = \mathcal{L}_{\text{critic}} + \mathcal{L}_{\text{actor}} + \mathcal{L}_\alpha \tag{1}$$

where, $\mathcal{L}_{\text{critic}}$, $\mathcal{L}_{\text{actor}}$ and $\mathcal{L}_\alpha$ represent the critic loss, actor loss, and temperature loss, respectively. For further details, please refer to Appendix B.2.

## 4 Methods

The proposed ProSpec method primarily consists of an FDM for incorporating PT into model-free RL, a prospective mechanism to prevent the entrapment in state traps, and cyclical consistency to improve data efficiency (see Figure 2 for schematic diagram). This section provides a detailed description of the ProSpec framework. For convenience, we provide a list of all the symbols used throughout the paper in the Appendix A.

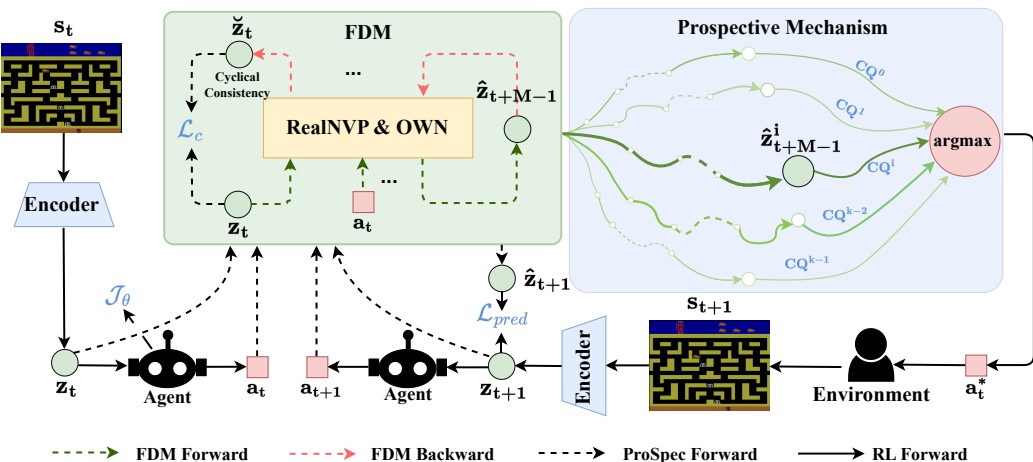

Figure 2: Training Procedure of ProSpec. Here, $\mathcal{J}_\theta$ represents the RL loss; $\mathcal{L}_{pred}$ denotes the prediction loss of the FDM; $\mathcal{L}_c$ stands for the cyclical consistency loss.

## 4.1 INCORPORATING PROSPECTIVE THINKING BY FDM

Currently, model-free methods lack prospective prediction capabilities and typically rely on blind trial and error, which significantly impacts decision-making. To address this, we introduce a dynamics model to incorporate PT into model-free RL, which predicts future states and enhance feature representation learning by generating a large number of virtual trajectories. Unlike the PlayVirtual (Yu et al., 2021), which uses forward and backward models for curiosity exploration. ProSpec achieves bidirectional inference through a single model, which predicts future states in the forward and backward to past states through action interventions. Specifically, we adopt a flow-based neural network architecture, Real-valued Non-Volume Preserving (RealNVP) (Dinh et al., 2022). RealNVP operates through bijective coupling layers, where each layer processes different parts of the input: $a_t \in \mathbb{R}^{(1:d) \times D}$ and $z_t \in \mathbb{R}^{(d+1:D) \times D}$(for ease of reading, we will simplify it to $a_t \in \mathbb{R}^{1:d}$ and $z_t \in \mathbb{R}^{d+1:D}$). RealNVP performs state transformations through scaling transformations $\exp(sc_i(\cdot))$ , translation transformations $t_i(\cdot)$ (implemented via MLP), and element-wise multiplication '$\odot$' and addition '+'. As shown in Figure 2, the encoder $f(\cdot)$ first converts image-based observations/states $s_t$ into latent representations $z_t = f(s_t)$. Then, predict the future latent state $\hat{z}_{t+1}$ based on the action $a_t$. Formally:

$$\hat{z}_{t+1} = \begin{cases} z_0 = f(s_0) & , t = 0 \\ h(\hat{z}_t, \hat{a}_t) & , t > 0, \hat{a}_t \sim \pi \end{cases} \tag{2}$$

where, the prospective prediction $h(\cdot, \cdot)$ is expressed as (more details can be seen in the Appendix C.3):

$$h(\hat{z}_t, \hat{a}_t) = \begin{cases} \hat{z}_{t+1}^{1:d} = \hat{a}_t \odot \exp(sc_2(\hat{z}_t)) + t_2(\hat{z}_t) \\ \hat{z}_{t+1}^{d+1:D} = \hat{z}_t \odot \exp(sc_1(\hat{z}_{t+1}^{1:d})) + t_1(\hat{z}_{t+1}^{1:d}) \end{cases} \tag{3}$$

Given a period of time $U$, the optimization objective of $h(\cdot, \cdot)$ is to minimize the difference (error) between the predicted latent state $\hat{z}_t$ and the latent state $z_t$ directly extracted from the raw observation/state. Following the SPR, we use cosine similarity to compute the prediction error in the DMControl environment (with specific details of the Atari benchmark provided in Appendix C.2). The optimization objective can be formalized as follows:

$$\mathcal{L}_{\text{pred}} = -\sum_{u=1}^{U} \left( \frac{\hat{z}_{t+u}}{\|\hat{z}_{t+u}\|_2} \right)^{\top} \left( \frac{z_{t+u}}{\|z_{t+u}\|_2} \right) \tag{4}$$

## 4.2 PROSPECTIVE MECHANISM TO PREVENT THE ENTRAPMENT IN STATE TRAPS

Model-free methods optimize the policy function through trial-and-error, demonstrating significant adaptability and advantages in continuous control tasks (Hansen et al., 2022). However, blindly

exploring the action space through trial-and-error to maximize cumulative rewards can lead the agent into dangerous state traps. To address this, we introduce PT into model-free methods for the first time, facilitating safe decision-making prior to execution and avoiding high-risk actions that could lead to "dead ends". Specifically, we implement the prospective mechanism using model prediction control (MPC) and value-consistency constraints. It is similar to MPC in classical control tasks, where the policy $\pi$ typically involves trajectory optimization to find the locally optimal solution at each time step (Hansen et al., 2022). The implementation of MPC involves predicting the optimal action sequence $\{a_t, a_{t+1}, \ldots, a_{t+N-1}\}$ over the next $N$ steps, and executing the first action $a_t$:

$$\pi_\theta^{\mathcal{MPC}} = \mathbb{E}[\sum_{j=0}^{N-1} \gamma^{t+j} \mathcal{R}(s_{t+j}, a_{t+j})] \tag{5}$$

Therefore, after the introduction of MPC, the agent selects the first action $a_t$ from a limited range $N$ to maximize the long-term reward over $j \in [0, N-1]$:

$$a_t = \underset{a_{t:t+N-1}}{arg\,max} \sum_{j=0}^{N-1} \gamma^{t+j} \mathcal{R}(s_{t+j}, a_{t+j}) \tag{6}$$

However, due to its deep understanding of the environment, MPC could autonomously compute the reward through virtual trajectories. This distinguishes it from the reward function in model-free methods, which is provided by the environment. Additionally, this differs from iterative Q-value estimation methods in RL, where MPC prioritizes short-term gains over long-term cumulative rewards. To achieve this, we propose imposing a value-consistency constraint on the "ahead plan" of the prospective mechanism. Specifically, the evaluation of actions in the prospective mechanism should align with the Q-values of the RL policy $\pi$, ensuring that at any given moment, the agent can select the action that provides the maximum long-term reward, even if the model has not yet fully converged. Formally, as shown in Figure 1, based on the imagined $k$-step action sequence $\hat{a}_0^{0:k-1} = \{\hat{a}_0^0, \cdots, \hat{a}_0^{k-1}\} \sim \pi(\cdot|z_0)$, we compute the cumulative discounted return for each trajectory $CQ_i = \sum_{j=0}^t \gamma^j Q(\hat{z}_i^j, \hat{a}_i^j)$ to determine the locally optimal action $a_0^* = \arg\max_{a_0}\{CQ_0, \ldots, CQ_{k-1}\}$.

Through the value-consistency constraint, we ensure that the action $a_t^*$ selected at each time step in the prospective mechanism is the optimal choice under the current policy, thereby enabling effective guiding future scenarios. Even in the worst case—where both the sampled actions are not optimal—this quasi-residual trial-and-error learning method allows the model-free methods to improve with each update, even if its worst learning outcome is merely an improvement based on its own trial-and-error policy. Appendix D further derives the intrinsic connection between value consistency constraints and policy learning consistency in model-free reinforcement learning, providing theoretical justification for our method.

In summary, employing a human-like prospective mechanism to "plan ahead, then execute" enhances safety by avoiding state trap risks and significantly improves data efficiency through more effective decision-making.

## 4.3 CYCLICAL CONSISTENCY FOR ENHANCING DATA EFFICIENCY

Previous studies have shown that applying cyclical consistency constraints in both forward and backward prediction can facilitate feature representation learning, thereby improving data efficiency (Yu et al., 2021). However, these methods typically require maintaining two separate dynamics models, which may be affected by bias, potentially leading the environment to fall into state traps. Additionally, training both models is challenging and costly. To address this issue, we propose the use of FDM to enable bidirectional computation. By employing appropriate action interventions, FDM can efficiently backtrack to previous states. Formally, given a subsequent state $\hat{z}_{t+M-1}$, we iteratively compute the previous state sequence $\breve{z}_{t+M-2:t} = \{\breve{z}_{t+M-2}, \cdots, \breve{z}_t\}$ using the backward function $h^{-1}(\cdot, \cdot)$:

$$\breve{z}_{t+M-2}, \hat{a}_{t+M-2} = h^{-1}(\hat{z}_{t+M-1}) \tag{7}$$

where, $h^{-1}(\cdot, \cdot)$ can be calculated as follows:

$$\breve{z}_t = \left(\breve{z}_{t+1}^{d+1:D} - t_1\left(\breve{z}_{t+1}^{1:d}\right)\right) \odot \exp\left(-s_1\left(\breve{z}_{t+1}^{1:d}\right)\right)$$
$$\hat{a}_t = \left(\breve{z}_{t+1}^{1:d} - t_2(\breve{z}_t)\right) \odot \exp(-s_2(\breve{z}_t)) \tag{8}$$

Note that there is a dimensional inconsistency between $\breve{z}_t \in \mathbb{R}^D$ and $z_t \in \mathbb{R}^{d+1:D}$. Typically, the linear transformation in a feed-forward neural network can be expressed as $\breve{z}_t' = W \times \breve{z}_t + B$, where $\breve{z}_t' \in \mathbb{R}^{d+1:D}$, $W \in \mathbb{R}^{d+1:D}$, and $B \in \mathbb{R}^{d+1:D}$ represent the weights and biases. However, due to the typical non-invertibility of $W$, backward inference (i.e., directly computing $\breve{z}_t = W^{-1} \times (\breve{z}_t' - B)$) presents challenges. To ensure reversibility, we employ Orthogonal Weight Normalization (OWN) (Huang et al., 2018), which enables bidirectional inference by enforcing orthogonality in the weights. As a result, the backward inference of $\breve{z}_t$ can be re-expressed as:

$$\breve{z}_t = W^T \times (\breve{z}_t' - B) \tag{9}$$

By combining RealNVP and OWN, we can effectively achieve bidirectional transformation between $\breve{z}_t$ and $\breve{z}_{t+1}$ (for more details, see Appendix C.3).

As illustrated in Figure 2, we introduce cyclical consistency constraints to facilitate feature representation learning. Ideally, if we have an encoder that can transform the observed state into appropriate features, the initial latent state $z_0$ of the cyclical trajectory composed of forward and backward trajectories should match the final latent state $\breve{z}_0$. Formally, given the latent state $\hat{z}_t$ at time $t$, the cyclical consistency loss is computed by calculating the distance error between $M$ pairs of virtual cyclical states $\hat{z}_{t+m}$ and $\breve{z}_{t+m}$:

$$\mathcal{L}_c = \frac{1}{M} \sum_{m=0}^{M-1} d(\hat{z}_{t+m}, \breve{z}_{t+m}) \tag{10}$$

The cyclical consistency constraint serves multiple purposes: 1) generate or sample a large number of virtual actions in an unsupervised manner to create diverse virtual trajectories, thus improving data efficiency and reducing reliance on real trajectories; 2) improving state reversibility and preventing the agent from falling into state traps; and 3) improving the training efficiency of the weight matrix $W$ in OWN.

## 4.4 OVERALL OBJECTIVE

As shown in Figure 2, the overall training objective of our approach can be summarized as follows:

$$\mathcal{L}_{total} = \mathcal{J}_\theta + \lambda_{pred}\mathcal{L}_{pred} + \lambda_c\mathcal{L}_c \tag{11}$$

where $\mathcal{J}_\theta$ is the loss function for RL (see section 3), and $\lambda_{pred}$ and $\lambda_c$ are hyperparameters that weight different losses (Appendix B shows more detail).

## 5 EXPERIMENTS

In this section, we provide a detailed description of the evaluation methods, including environment settings, experimental details, and evaluation metrics. Subsequently, we conduct a series of ablation experiments to analyze the effectiveness of the key components of our proposal.

### 5.1 SETUP FOR EVALUATION

**Environments.** We conducted benchmark tests on ProSpec in environments with limited interaction, using DMControl for continuous control testing (Tassa et al., 2018) and selecting Atari (Bellemare et al., 2013) for discrete control evaluation. Consistent with the practices of most researchers (Wang et al., 2016; Edwards et al., 2018; Van Hasselt et al., 2019; Vinyals et al., 2019; Yarats et al., 2020; 2021; Yu et al., 2021; Yue et al., 2023), we evaluated performance after training for 100k and 500k interaction steps in DMControl and after 100k interaction steps in Atari. In total, we selected 32 games from both environments for performance evaluation.

**Baseline.** Some recent excellent decision methods are selected as baselines, including Dreamer (Hafner et al., 2019), SLAC (Guo et al., 2020), CURL (Laskin et al., 2020a), DrQ (Yarats et al., 2020), SAC+AE (Yarats et al., 2021), and SPR (Schwarzer et al., 2021) in the DMControl benchmark. For Atari, we selected DER (Van Hasselt et al., 2019), OTR (Kielak, 2019), SimPLe (Kaiser et al., 2020), CURL (Laskin et al., 2020a), DrQ (Yarats et al., 2020) as baselines, as these are state-of-the-art methods in Atari according to their publications. Notably, the most recent methods, VCR (Yue et al., 2023) and PLASTIC (Lee et al., 2024) are evaluated on both DMControl and Atari, whereas RLTSC

Table 1: Scores obtained by different methods on DMControl-100k and DMControl-500k (mean and standard deviation). ProSpec was run with 10 random seeds, and the overall median and mean scores were recorded for evaluation comparison.

| | 100k Step Scores | | | | | | | | | |
| TASK | SLAC (2020) | CURL (2020) | SAC+AE (2020) | DrQ-V2 (2021) | SPR† (2021) | PlayVirtual (2021) | Dreamer-V3 (2022) | VCR (2023) | PLASTIC (2024) | ProSpec (Ours) |
|---|---|---|---|---|---|---|---|---|---|---|
| Finger, spin | 859±132 | 767±56 | 740±64 | 325±292 | 835±154 | 866±103 | 309.3±106 | 795±157 | 676±131 | **875±104** |
| Cartpole, swingup | 305±66 | 582±146 | 311±11 | 677±214 | 806±45 | 806±45 | 345.3±38 | 815±47 | 781±26 | **833±34** |
| Reacher, easy | 305±43 | 538±233 | 274±14 | 256±145 | 667±173 | 685±217 | 640.8±164 | 763±112 | **836±153** | 782±137 |
| Cheetah, run | 344±69 | 299±48 | 267±24 | 273±130 | 434±48 | 476±67 | 536.8±43 | 422±54 | 464±50 | **477±24** |
| Walker, walk | 541±98 | 403±24 | 394±22 | 171±160 | 398±145 | 596±184 | 614.6±146 | 650±143 | **710±120** | 507±60 |
| Ball in cup, catch | 756±314 | 769±43 | 391±82 | 359±228 | 814±263 | 921±31 | 368±64 | 858±85 | 921±36 | **927±17** |
| Median Score | 422.5 | 560 | 351 | 299 | 736.5 | 745.5 | 713.4 | 779 | 791 | **807.5** |
| Mean Score | 518.3 | 559.7 | 396.2 | 348 | 659 | 725 | 698.5 | 717.2 | 731.3 | **733.5** |
| | 500k Step Scores | | | | | | | | | |
| Finger, spin | 673±92 | 926±45 | 884±128 | 789±124 | 924±132 | 963±40 | 818.5±135 | 972±25 | 969±21 | **973±14** |
| Cartpole, swingup | 764±43 | 841±45 | 735±63 | 845±18 | 870±12 | 865±11 | 819.1±12 | 854±26 | 864±15 | **871±13** |
| Reacher, easy | 628±94 | 929±44 | 627±58 | 748±229 | 925±79 | 942±66 | 898.9±48 | 938±37 | 947±24 | **965±14** |
| Cheetah, run | 640±19 | 518±28 | 550±34 | 607±32 | 734±45 | 719±51 | 728.7±69 | 661±32 | **746±55** | 714±57 |
| Walker, walk | 842±51 | 902±43 | 847±48 | 696±370 | 916±75 | 928±30 | 955.8±14 | 930±18 | **961±26** | 954±19 |
| Ball in cup, catch | 852±71 | 959±27 | 794±58 | 844±174 | 963±8 | 967±5 | 957.1±7 | 958±4 | 936±8 | **972±5** |
| Median Score | 718.5 | 914 | 764.5 | 768.5 | 920 | 935 | 858.7 | 934 | 941.1 | **959.5** |
| Mean Score | 733.6 | 779.2 | 739.5 | 754.8 | 885.7 | 897.3 | 863 | 885.5 | 903.8 | **908.2** |

(Zheng & Song, 2024) is evaluated only on Atari. Additionally, we selected PlayVirtual (Yu et al., 2021) as a strong baseline. It is important to note that since SPR was originally designed for offline tasks, we adapted it for continuous tasks using a SAC-based approach (SPR†).

**Implementation Details.** In DMControl, we adopted the encoder and policy network architecture from CURL (Laskin et al., 2020a), removing the contrastive loss component and introducing BYOL (Grill et al., 2020) to establish a baseline similar to SPR. Following PlayVirtual, the number of cyclical consistency steps is set to 6, with actions sampled randomly from a uniform distribution over the continuous action space. According to the ablation experiment (Appendix F.3), the prospective count $k$ was set to 3, and the prediction horizon $t$ was set to 6. In Atari, our ProSpec solution was based on the official SPR and PlayVirtual code, with the cyclical consistency steps also set to 9 for generating imaginary trajectories. The prospective count $k$ is set to 9, and the prediction horizon $t$ is set to 3. For DMControl-100k, we reran the official SPR, PlayVirtual, and PLASTIC code with 10 seeds. To reduce variance, we performed evaluations with 10 random seeds for each games. Given the limitations in computational resources, we conduct the ablation experiments using 5 random seeds. We did not rerun VCR because the official code was not available. Due to computational resource limitations, follow other work in the community(Schwarzer et al., 2021; Yu et al., 2021; Yue et al., 2023), we report the DMControl-500k and Atari results directly from the respective papers. All implementations were carried out using PyTorch (Paszke et al., 2019), with network training performed on an NVIDIA RTX 3090 GPU. Further details can be found in Appendix E.

**Evaluation Metrics.** In DMControl, the maximum achievable score for each environment is capped at 1000 (Tassa et al., 2018). We evaluated model performance across six common environments, using the median score as the overall performance metric (Laskin et al., 2020a; Guo et al., 2020; Yarats et al., 2021). For Atari discrete action tasks, we used the interquartile mean (IQM) Human Normalized Score (HNS) as the primary metric, following VCR.

## 5.2 Performance Comparison with SOTA

**Comparison on DMControl.** As shown in Table 1, our method outperformed most other approaches, achieving four top scores across six environments, even with limited interactions (100k or 500k). Specifically, when considering the median scores, our method excelled in two key aspects: (i) In the DMControl-100k with limited interaction data, our method achieved the highest median score of 807.5, surpassing PLASTIC by 2.1%, PlayVirtual by 8.32%, VCR by 3.66%, SPR by 9.64%, DrQ by 17.80%, and CURL by 44.20%. (ii) In the DMControl-500k, our method achieved a median score of 959.5 across five environments, which is very close to the maximum achievable score of 1000. These results suggest that tasks in the DMControl often involve dynamic goals (such as maintaining balance or reaching a specific speed), where PT plays a crucial role. For example, in the *Cheetah, run* task, PT helps the agent predict the long-term impact of the next action on its running speed and gait,

Table 2: Scores obtained by different methods on Atari with 100k interaction steps. ProSpec was run with 10 random seeds, and the overall IQM HNS scores were recorded for evaluation comparison. Due to space limitations, only the results with the highest performance are presented here, and the complete results can be found in the Appendix Table 5.

| Game | Human (-) | Random (-) | DER (2019) | OTR (2019) | SimPLe (2020) | CURL (2020) | DrQ-V2 (2021) | SPR (2021) | PlayVirtual (2022) | VCR (2023) | PLASTIC (2024) | RLTSC (2024) | ProSpec (Ours) |
|---|---|---|---|---|---|---|---|---|---|---|---|---|---|
| Bank Heist | 753.1 | 14.2 | 51 | 182.1 | 34.2 | 131.6 | 168.9 | 380.9 | 245.9 | 303.7 | 161.2 | 473.6 | **481.1** |
| Battle Zone | 37,187.5 | 2,360 | 10,124.6 | 4,060.6 | 5,184.4 | 1,870 | 12,954 | 16,651 | 13,260 | 13,261 | 2,099 | 11,700 | **15,934.4** |
| Chopper Command | 7,387.8 | 811 | 861.8 | 1,033.3 | 1,246.9 | 1,058.5 | 780.3 | 974.8 | 974.8 | 1,024.2 | 891.2 | 1,120.6 | **1,351.0** |
| Gopher | 2,412.5 | 257.6 | 349.5 | 778 | 771 | 669.3 | 636.3 | 715.2 | 684.3 | 539.7 | 839.4 | 682.4 | **866** |
| Kung Fu Master | 22,736.3 | 258.5 | 14,346.1 | 5,722.2 | 17,257.2 | 14,307.8 | 9,111 | 13,192.7 | 14,259 | 19,679.7 | 16,105 | 12,680.4 | **20,711** |
| Private Eye | 69,571.3 | 24.9 | 72.8 | 59.6 | 34.9 | 81.9 | 3.5 | 86 | 93.9 | 98.9 | **100** | **100** | **100** |
| Seaquest | 42,054.7 | 68.4 | 354.1 | 286.9 | 683.3 | 384.5 | 301.2 | 583.1 | 521.2 | 635.9 | 527.7 | 396.8 | **800.4** |
| IQM HNS (%) | - | - | 14.2 | 18.6 | 23.9 | 18.9 | 24.5 | 45.0 | 43.4 | 46.4 | 42.1 | 43.3 | **47.2** |

Table 3: Effectiveness of (a) Prospevtive Mechanism &(b) Cyclical Consistency &(c) FDM on ProSpec.

| (a) Prospevtive Mechanism | | | (b) Cyclical Consistency | | | (c) Dynamic Model | | |
|---|---|---|---|---|---|---|---|---|
| | DMControl | Atari | | DMControl | Atari | | DMControl | Atari |
| SPR$^\dagger$ | 736.5 | 45.0 | SPR$^\dagger$ | 736.5 | 45.0 | SPR$^\dagger$ | 736.5 | 45.0 |
| PlayVirtual | 745.5 | 43.4 | PlayVirtual | 745.5 | 43.4 | PlayVirtual | 745.5 | 43.4 |
| ProSpec-NP | 752.0 | 43.5 | ProSpec-NC | 752.8 | 45.4 | ProSpec-MLP | 773.1 | 44.8 |
| **ProSpec** | **807.5** | **47.2** | **ProSpec** | **807.5** | **47.2** | **ProSpec** | **807.5** | **47.2** |

thereby avoiding premature or delayed adjustments. Overall, our method demonstrates exceptional performance, even with limited data, and also excels in terms of asymptotic performance.

**Comparison on Atari.** Table 2 presents a comparison with state-of-the-art methods, along with results for expert human and random play, as reproduced from (Yarats et al., 2020). ProSpec achieves an IQM HNS of 47.2%, outperforming various methods: 97.4% higher than SimPLe, 232.4% higher than DER, 153.7% higher than OTR and CURL, 92.6% higher than DrQ, 4.8% higher than SPR, 1.7% higher than VCR, 8.7% higher than PlayVirtual, 12.1% higher than PLASTIC, and 9.0% higher than RLTSC. In terms of individual games, ProSpec achieved the highest score in 7 out of 26 game scenarios. Notably, it demonstrated performance comparable to that of expert humans in certain environments, particularly in the *Bank Heist* scenario. In this scenario, ProSpec's score of 481.1 significantly outperforms other state-of-the-art models and approaches the expert human score of 753.1. The objective in *Bank Heist* is to evade the police and rob as many banks as possible, requiring the agent to predict the police's potential locations and actions to optimize the robbery route and avoid capture. Our method's emphasis on PT enables preemptive action planning, contributing to its superior performance over other models that lack this capability. The complete set of results, refer to Appendix Table 5.

**Analysis of ProSpec.** Through extensive experiments, we identify three domains in which ProSpec consistently outperforms baseline methods: **Deep prospective planning:** In games like *Bank Heist*, *Battle Zone*, and *Chopper Command*, success relies on forecasting patrols, optimizing multi-step attacks, and avoiding collisions. ProSpec's FDM generates candidate trajectories and picks the locally optimal, highest-reward action—outperforming pure trial-and-error on long-horizon tasks. **High-risk trap avoidance:** Titles such as *Chopper Command* and *Private Eye* end upon any mistake. By simulating ahead, ProSpec detects and avoids dangerous states, reducing resets and improving safety under risk. **Sparse-reward, goal-driven tasks:** In *Amidar* and *Kung Fu Master*, infrequent rewards make naive exploration slow. ProSpec's forecasting supplements sparse feedback identifies promising strategies, and speeds convergence to near-optimal performance. In contrast, in dense-reward, simple environments (e.g., *Freeway*), trial-and-error already performs well, and ProSpec's lookahead adds only modest gains (see Appendix F.2 for more analysis)

## 5.3 ABLATIONS

In this section, we present ablation studies on the DMControl-100k and Atari environments to investigate the key components of ProSpec. Due to space constraints, some of the ablation results are provided in the Appendix F.3. Given the limitations in computational resources, we conduct the experiments using 5 random seeds.

**Effectiveness of Prospective Mechanism.** To explore the effect of prospective mechanism on RL tasks, we evaluated ProSpec without the prospective mechanism (ProSpec-NP, i.e., the model-free baseline policy) and compared it with SPR[†] and PlayVirtual. As can be seen in Table 3 (a): (i) In DMControl, ProSpec improved its median score by 55.5 points compared to ProSpec-NP, and in Atari, it improved the IQM HNS score by 8.5%; (ii) ProSpec-NP outperformed PlayVirtual by 6.5 median points and 0.1% IQM HNS score across both benchmarks; (iii) ProSpec increased PlayVirtual's median score from 745.5 to 807.5, and its IQM HNS score from 43.4% to 47.2%.

These results can be attributed to several factors. First, the FDM incorporates PT into model-free RL, enabling it to predict future n-stream trajectories and select higher-value, lower-risk decisions. This leads to performance improvements of 55.5 and 62 points, along with IQM HNS gains of 8.5% and 8.7%, compared to ProSpec-NP and PlayVirtual, respectively. Second, the proposed prospective mechanism enables ProSpec to execute decisions only after ahead planning, thereby avoiding high-risk actions that may lead to "dead ends," which allows ProSpec to outperform SPR[†].

**The Impact of Cyclical Consistency on ProSpec.** As described in Section 4.3, cyclical consistency addresses a fundamental challenges in RL: generating a large number of virtual trajectories to improve data efficiency through augmented state feature representations. We evaluated the performance of ProSpec without the cyclical consistency constraints (referred to as ProSpec-NC) and compared it to SPR[†] and PlayVirtual. As shown in Table 3(b), regularizing the encoder with $\mathcal{L}_c$ in ProSpec-NC resulted in a gain of 7.3 and 4.0%. In contrast, regularizing both the encoder and the FDM resulted in a gain of 62 and 8.8%. It is clear that cyclical consistency can benefit the training of the FDM, but the greater benefit comes from the learning of the feature representations in the encoder (Yu et al., 2021).

**The Effectiveness of Flow-Based Dynamic Model.** We also investigated the impact of different dynamical models on the performance of ProSpec. As shown in Table 3(c), flow-based dynamic model significantly enhanced performance, achieving scores of 807.5 and 47.2 on the DMControl and Atari benchmarks, respectively. Compared to SPR[†], this resulted in improvements of 71 and 4.8%, while compared to PlayVirtual, it improved by 62 and 9.4%. Compared to ProSpec-MLP, which employs an MLP as the dynamical model, performance improved by 34.4 and 5.4%, respectively. This result can be explained as follows: (i) FDM provides more accurate backward predictions, and cyclical consistency enhances state reversibility and guides low-risk decision-making, enabling ProSpec to outperform ProSpec-MLP on both benchmarks. (ii) Leveraging prospective mechanism, ProSpec-MLP and ProSpec analyze situations from multiple prospectives to make better decisions, thereby avoiding state traps and outperforming PlayVirtual. (iii) Self-supervised methods enhance data efficiency by generating multiple state-action trajectories, allowing ProSpec to outperform SPR[†].

**The Impact of Prospective Decision Learning on Data Efficiency.** We also evaluated the impact of prospective decision-making on data efficiency using DMControl and Atari, comparing ProSpec against ProSpec-NP. As illustrated in Fig. 5-8 (Appendix F.4), ProSpec consistently achieves higher rewards with fewer training steps, demonstrating markedly superior efficiency. These findings show that prospective decision-making promotes safer, reward-oriented action selection, yielding smoother learning curves and more stable performance across environments. Comprehensive results on data efficiency and time complexity are reported in Appendix F.4 and Appendix F.5.

## 6 CONCLUSION

This paper proposes a novel ProSpec RL method, the first to incorporate prospective decision learning to model-free RL for efficient and safe exploration. Firstly, we propose a flow-based reversible dynamics model to incorporate PT into model-free RL by predicting future n-stream trajectories based on the current state and policy. Meanwhile, we propose a prospective mechanism to prevent the entrapment in state traps, which uses model predictive control with value consistency constraint to enable the learning to plan ahead then execute, avoiding "dead ends" caused by high-risk actions. Additionally, we present a cyclical consistency constraint to improve data efficiency by generating accurate and reversible virtual trajectories to further enhance state feature representations. Comprehensive evaluations of ProSpec on DMControl and Atari benchmarks demonstrate the significant accelerations in the model decision learning and the state-of-the-art performance in 4 of 6 DMControl and 7 of 26 Atari games.

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

APPENDIX

CONTENTS

## A  SYMBOLS DEFINITION

For clarity, we have listed all the symbols used throughout the paper and their meanings in Table 4.

Table 4: Definitions of the symbols used in this document.

| Symbol | Definition |
|--------|-----------|
| $\mathcal{S}$ | the set of states |
| $\mathcal{A}$ | action space |
| $\mathcal{T}(\cdot, \cdot)$ | dynamic function |
| $\mathcal{R}(\cdot, \cdot)$ | reward function |
| $\gamma$ | discount factor |
| $G_t(\cdot)$ | cumulative discounted return |
| $\mathcal{Q}_\pi(\cdot)$ | action-value function |
| $\pi^*$ | optimal policy |
| $a$ | action |
| $s$ | state |
| $\tilde{z}$ | latent state |
| $\hat{z}$ | dynamic model forward predicts latent state |
| $\check{z}$ | dynamic model backward predicts latent state |
| $sc(\cdot)$ | scaling function |
| $t(\cdot)$ | translation function |
| $CQ$ | cumulative discounted value |
| $\mathcal{O}$ | matrix family |
| $k$ | the count of prospective |
| $t$ | prediction horizon (time) |
| $N$ | the range of time (MPC) |
| $U$ | total time in forward prediction loss |
| $W$ | weight |
| $I$ | identity matrix |
| $B$ | bias |
| $M$ | number of sampled actions in cyclical consistency loss |
| $h(\cdot, \cdot)$ | FDM forward calculation function |
| $h^{-1}(\cdot, \cdot)$ | FDM backward calculation function |
| $q(\cdot)$ | prediction head |
| $g(\cdot)$ | online projection heads |
| $g_m(\cdot)$ | target projection heads |
| $d(\cdot, \cdot)$ | distance metrics |

## B  ALGORITHM

### B.1  SOFT ACTOR CRITIC

Soft Actor-Critic (SAC) (Haarnoja et al., 2018) is a widely used gradient-based algorithm for continuous control that incorporates policy entropy as an additional reward to encourage exploration. Let $\theta$ denote the parameters of the stochastic policy $\pi$ and $\phi$ denote the parameters of $Q(s_t, a_t)$. SAC tries to learn a probabilistic policy $\pi_\theta$, two Q functions $Q_{\phi_1}$ and $Q_{\phi_2}$, and a temperature parameter $\alpha$ to adjust the balance between exploration and exploitation.

**Critic.** To mitigate the risk of overestimation, SAC adopts a twin Q-network structure and chooses the minimum estimate from the two Q-functions. SAC constructs the target loss function for Critic updates using transitions sampled from the experience replay buffer $(s_t, a_t, s_{t+1}, r_t, done)$:

$$\mathcal{L}_\phi^{\text{critic}} = \sum_{i=1,2} (Q_{\phi_i}(s_t, a_t) - y(r_t, s_{t+1}, done))^2 \tag{12}$$

where $r_t$ denotes the reward, $s_{t+1}$ represents the next state, and $done$ is the termination flag, with a value of 1 indicating that $s_{t+1}$ is a terminal state and 0 otherwise. The target is the clipped gradient, defined as:

$$y(r_t, s_{t+1}, done) = r_t + \gamma(1 - done)(\min_{i=1,2} Q_{\phi_{target,i}}$$
$$(s_{t+1}, a_{t+1} - \alpha \log_{\phi_\theta}(a_{t+1}|s_{t+1}))) \tag{13}$$

where $\gamma$ is the discount factor and $Q_{\phi_{target,i}}$ is the target Q-network updated using the exponential moving average (EMA) of $Q_{\phi_i}$.

**Actor.** The objective of the actor $\pi_\theta$ is to select actions that maximize the action value and policy entropy. Therefore, the update loss function for the actor can be defined as:

$$\mathcal{L}_\theta^{\text{actor}} = - \left( \min_{i=1,2} Q_{\phi_i}\left(s_t, a_\theta(s_t)\right) - \alpha \log \pi_\theta\left(a_\theta(s_t) \mid s_t\right) \right) \tag{14}$$

where $a_\theta(s_t)$ is sampled from a stochastic policy $\pi_\theta(s_t)$.

**The overall objective.** The overall training objective of SAC is as follows:

$$\mathcal{J}_\theta^{\mathcal{SAC}} = \mathcal{L}_\phi^{\text{critic}} + \mathcal{L}_\theta^{\text{actor}} + \mathcal{L}_\alpha \tag{15}$$

where, $\mathcal{L}_\phi^{\text{critic}}$, $\mathcal{L}_\theta^{\text{actor}}$ and $\mathcal{L}_\alpha$ represent the critic loss, actor loss and temperature loss, respectively.

### B.2  DQN

The Deep Q-Network (DQN) (Mnih et al., 2015) encodes the state through a neural network and outputs the corresponding Q-values for the actions, thereby enabling Q-learning to scale to large state spaces. Additionally, DQN incorporates techniques such as experience replay and target network separation to stabilize the training process. The overall objective of DQN is represented by the following loss function:

$$\mathcal{J}_\theta^{\mathcal{DQN}} = \mathbb{E}_{(s,a,s',r)\sim\mathcal{D}}[(r + \gamma \max_{a'} Q_{\bar{\theta}})(s', a') - Q_\theta(s, a))^2] \tag{16}$$

Where, $Q_\theta$ denotes the Q network controlled by parameters $\theta$, and $Q_{\bar{\theta}}$ denotes the target Q network. Meanwhile, $\mathcal{D}$ denotes the replay buffer storing the experience tuples.

### B.3  PSEUDO CODE

Here we present the pseudo-code for ProSpec, which can be seamlessly integrated into value-based algorithms. Specifically, Algorithm 1 outlines the prospective prediction process of ProSpec, while Algorithm 2 describes the training process. For simplicity, the pseudo-code omits many details.

---

**Algorithm 1:** The prospective prediction process of ProSpec.

---

Denote parameters of encoder $f$, flow-based dynamic model $h$, value head $Q$ as $\theta$;
Denote the number of prediction horizon as $t$, the count of prospectives as $k$;
**Input:** current state $s_t$;
**if** *prospective* **then**
    Encoded into latent state representations as $z_t = f(s_t)$ by the encoder
    **for** *int i=0 to k* **do**
        $\hat{a}_t^i \sim \pi(z_t)$                                         ▷ prospectives counts $k$
        **for** *int j = 0 to t* **do**
            **if** $j > 0$ **then**
                $\hat{a}_{t+j} \sim \pi(\hat{z}_{t+j})$
            **else**
                $\hat{a}_{t+j} = \hat{a}_t^i$
            **end**
            $\hat{z}_{t+j} \leftarrow h(z_t, \hat{a}_{t+j})$                    ▷ prediction horizon $t$
            calculate $Q(\hat{z}_{t+j}, \hat{a}_{t+j})$
        **end**
        $CQ_i = \sum_{j=0}^t \gamma^j Q(\hat{z}_j, \hat{a}_{t+j})$             ▷ cumulative discounted return
    **end**
    $a_t^* = \arg\max_{a_0} \{CQ_1, \cdots, CQ_k\}$
**end**
**Output:** $a_t^*$;

---

## C  MORE IMPLEMENTATION DETAIL

### C.1  NETWORK ARCHITECTURE FOR DMCONTROL

Considering that the original SPR was designed for discrete control tasks, we have developed an SPR-like scheme, called SPR$^\dagger$, for continuous control tasks, following previous research (Yu et al., 2021; Yue et al., 2023). Specifically, we use the encoder and policy networks of CURL (Laskin et al., 2020a) as our base networks. Building on SPR (Schwarzer et al., 2021), we remove the contrast loss from CURL and introduce a BYOL (Grill et al., 2020) head to construct our SPR-like baseline scheme. For the dynamics model (DM) within SPR$^\dagger$, we adopt a network structure similar to that used in DBC (Zhang et al., 2021). The DM consists of two fully connected layers, with a linear normalization (LN) layer and a ReLU activation following the first fully connected layer. The encoder consists of four convolutional layers (each followed by a ReLU), followed by a fully connected layer, an LN layer, and a hyperbolic tangent (tanh) activation. Similar to the design in SPR, we use a projection head $g(\cdot)$, a prediction head $q(\cdot)$ for the online encoder, an impulse encoder $f_m(\cdot)$, and an impulse projection head $g_m(\cdot)$. Both the projection head and the prediction head consist of two fully connected layers (preceded by a ReLU layer), each containing 512 hidden units.

---

**Algorithm 2:** The training process of ProSpec.

---

Denote parameters of encoder $f$, flow-based dynamic model $h$, value head $Q$ as $\theta$;
Denote the number of prediction step as $t$, the number of viewing angles as $k$;
Denote the number of virtual trajectories as $M$;
Denote the prediction and cycle consistency loss weight as $\lambda_{pred}$ and $\lambda_c$;
Denote the replay buffer as $\mathcal{D}$;
Randomly initialize all network parameters and make the reply buffer empty.
**while** *train* **do**
    interact with the environment using the optimal action $a^*$
    record/collect experience $\mathcal{D} \leftarrow \mathcal{D} \cup (s, a*, s_{next}, r)$
    sample a aequence of $(s, a*, s_{next}, r) \sim \mathcal{D}$
    initialize $\mathcal{J}_\theta$, $\mathcal{L}_{pred}$ and $\mathcal{L}_c$ with 0
    $z_0 \leftarrow f(s_0)$
    **for** *u = 1 to M* **do**
        $\{\hat{a}_0^{(j)}, \hat{a}_1^{(j)}, \cdots, \hat{a}_{t-1}^{(j)}, \} \sim \mathcal{A}$         ▷ randomly sample a sequence of actions
        $\hat{z}_0^{(j)} \leftarrow z_0$
        **for** *i=0 to t* **do**
            $\hat{z}_{i+1}^{(j)} \leftarrow h(\hat{z}_i^{(j)}, \hat{a}_i^{(j)})$         ▷ forward prediction
        **end**
        $\check{z}_t^{(j)} \leftarrow \hat{z}_t^{(j)}$
        **for** *i=t-1 to 0* **do**
            $\hat{z}_i^{(j)} \leftarrow h^{-1}(\hat{z}_{i+1}^{(j)}, \hat{a}_i^{(j)})$         ▷ backward return
        **end**
        $\mathcal{L}_c \leftarrow \mathcal{L}_c/M$         ▷ cyclical consistency loss
    **end**
    using eq. 4 calculated $\mathcal{L}_{pred}$
    using eq. 1 calculated $\mathcal{J}_\theta$
    $\mathcal{L}_{total} \leftarrow \mathcal{J}_\theta + \lambda_{pred}\mathcal{L}_{pred} + \lambda_c\mathcal{L}_c$
    $\theta \leftarrow Optimize(\theta, \mathcal{L}_{total})$
**end**

---

### C.2  PROSPECTIVE PREDICTION IN ATARI

In the MDP framework, a one-step transition $(s_t, a_t, s_{t+1})$ describes the process of transitioning from the current state $s_t \in \mathcal{S}$ to the next state $s_{t+1}$ by taking action $a_t \in \mathcal{A}$. Considering the crucial aspect of PT, i.e., the ability to predict future scenarios, is very important for RL tasks. Many studies focus on training dynamic models to predict future states to learn good feature representations (Schwarzer

et al., 2021; Guo et al., 2020; Lee et al., 2020). In this study, the FDM $h(\cdot, \cdot)$ is used to simulate the dynamics function $\mathcal{T}$ in the latent embedding space, allowing us to explore the future from multiple prospectives. As shown in Figure 2, the encoder $f(\cdot)$ first converts the image-based observation/state $s_0$ into a latent representation $z_0 = f(s_0)$. Then, based on the predicted action $a_0$, the FDM predicts the latent state $\hat{z}_1$ for the future step. Formally:

$$\hat{z}_t = \begin{cases} z_0 = f(s_0) & , t = 0 \\ h(\hat{z}_{t-1}, \hat{a}_{t-1}) & , t > 0, \hat{a} \sim \pi \end{cases} \tag{17}$$

The optimization objective of the FDM $h(\cdot, \cdot)$ is to minimize the difference (error) between the predicted latent state $\hat{z}_t$ and the latent state $z_t$ directly extracted from the original observation/state. Following PlayVirtual, we use "projection" to predict the error on Atari to achieve the following objectives:

$$\mathcal{L}_{\text{pred}} = -\sum_{u=1}^{U} d\left(\hat{z}_{t+u}, \breve{z}_{t+u}\right) \tag{18}$$

where $d(\hat{z}_{t+u}, \breve{z}_{t+u})$ is distance metric for the latent state.

### C.3 THE FLOW-BASED DYNAMICS MODEL (FDM)

The FDM predicts the future and retroactively infers previous latent states, crucial in ProSpec. Unlike PlayVirtual (Yu et al., 2021), which employs separate models for these tasks, our approach employs a single model for both, avoiding training instability. We utilize Real-valued Non-Volume Preserving (RealNVP) (Dinh et al., 2022), a flow-based neural network that performs bijective coupling transformations, which can be seen in the figure 3a. RealNVP operates via bijective coupling layers, with each layer handling parts of the input $a_t \in \mathbb{R}^{1:d}$ and $z_t \in \mathbb{R}^{d+1:D}$ separately. It applies transformations using affine functions for scaling $\exp(sc_i(\cdot))$ and translation $t_i(\cdot)$, along with element-wise multiplication '$\odot$' and addition '+' operations. This enables the computation of the forward process as:

$$\begin{aligned} \hat{z}_{t+1}^{1:d} &= \hat{a}_t \odot \exp(sc_2(\hat{z}_t)) + t_2(\hat{z}_t) \\ \hat{z}_{t+1}^{d+1:D} &= \hat{z}_t \odot \exp(sc_1(\hat{z}_{t+1}^{1:d})) + t_1(\hat{z}_{t+1}^{1:d}) \end{aligned} \tag{19}$$

and the backward processes can be calculated as:

$$\begin{aligned} \breve{z}_t &= \left(\breve{z}_{t+1}^{d+1:D} - t_1\left(\breve{z}_{t+1}^{1:d}\right)\right) \odot \exp\left(-s_1\left(\breve{z}_{t+1}^{1:d}\right)\right) \\ \hat{a}_t &= \left(\breve{z}_{t+1}^{1:d} - t_2(\breve{z}_t)\right) \odot \exp(-s_2(\breve{z}_t)) \end{aligned} \tag{20}$$

As shown in the figure 3a, there is a dimensional mismatch between the quantities $\hat{z}_t \in \mathbb{R}^D$ and $\hat{z}_t' \in \mathbb{R}^{d+1:D}$, $\breve{z}_{t+1}' \in \mathbb{R}^{d+1:D}$ and $\breve{z}_{t+1} \in \mathbb{R}^D$, computed by RealNVP. Typically, a linear transformation of a feed-forward neural network can be expressed as:

$$\begin{aligned} \hat{z}_t' &= W \times \hat{z}_t + B \\ \breve{z}_{t+1}' &= W \times \breve{z}_{t+1} + B \end{aligned} \tag{21}$$

However, due to the typical non-invertibility of $W$, backward inference (i.e., directly computing $\breve{z}_{t+1} = W^{-1} \times (\breve{z}_{t+1}' - B)$) presents challenges. To ensure reversibility, we employ Orthogonal Weight Normalization (OWN) (Huang et al., 2018), which enables bidirectional inference by enforcing orthogonality in the weights. As a result, the backward inference of $\breve{z}_t$ can be re-expressed as:

$$\begin{aligned} \hat{z}_t &= W^T \times (\hat{z}_t' - B) \\ \breve{z}_{t+1} &= W^T \times (\breve{z}_{t+1}' - B), WW^T \in \mathcal{O}^{D \times D} \end{aligned} \tag{22}$$

where the matrix family $\mathcal{O}^{D \times D} = \{W \in \mathbb{R}^{(d+1:D) \times D} \mid WW^T = W^{-1}W = I\}$ (see (Huang et al., 2018) for more details). For clarity, we have summarized the reasoning process of OWN in Figure 3b. By using the combination of RealNVP and OWN, we can effectively achieve bidirectional transformation between $\breve{z}_t$ and $\breve{z}_{t+1}$.

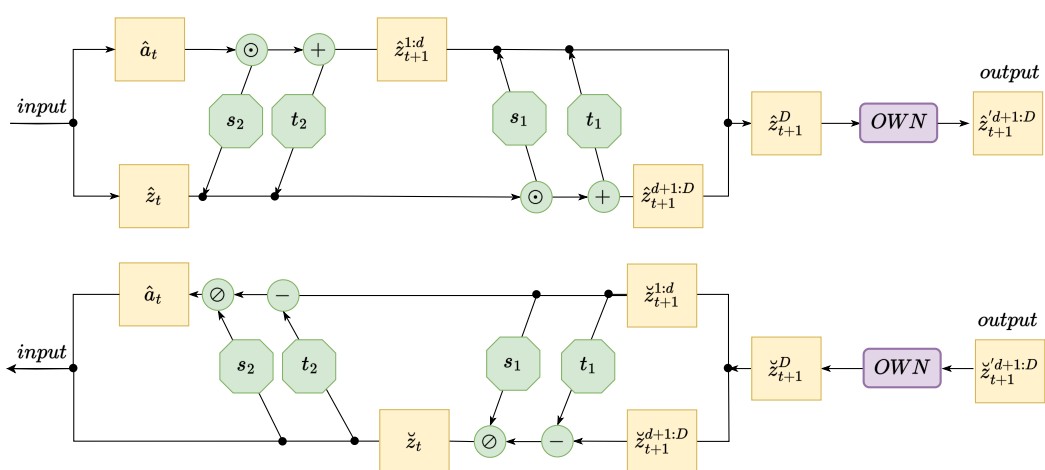

(a) The forward and reverse propagation process of RealNVP.

$$a_t \in \mathbb{R}^{(1:d) \times D}, \hat{z}_{t+1} \in \mathbb{R}^{D \times D}, W \in \mathbb{R}^{D \times (d+1:D)}, B \in \mathbb{R}^{D \times (d+1:D)}$$

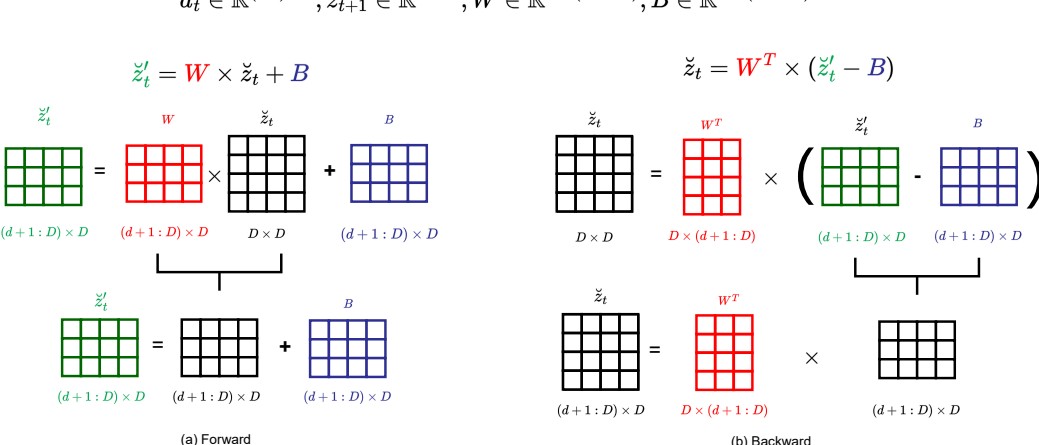

(b) The reasoning process of OWN.

Figure 3: Flow-based Reversible Dynamics Model of ProSpec.

## D    DERIVATION OF VALUE CONSISTENCY CONSTRAINT VIA CUMULATIVE Q-VALUE SCORING

### D.1    CUMULATIVE Q-VALUE TRAJECTORY SCORING

In traditional MPC frameworks, the trajectory score is defined by the discounted return over a finite horizon:

$$J(\tau) = \sum_{t=0}^{H} \gamma^t r_t \quad \text{or} \quad J(\tau) = \sum_{t=0}^{H} \gamma^t r_t + \gamma^H V(s_H), \tag{23}$$

where $H$ is the planning horizon, $\gamma \in (0, 1)$ is the discount factor, and $V(s_H)$ denotes the terminal value function. However, this mechanism faces three critical challenges in practice:

- **Short-horizon bias**: Planning focuses only on immediate rewards within a limited horizon, neglecting long-term potential.
- **Sparse reward failure**: In sparse reward settings, different trajectories tend to yield similar cumulative returns, making it difficult to distinguish their quality.
- **Insufficient state resolution**: In high-frequency control tasks (e.g., robotic control with a sampling interval of 0.5 ms), consecutive states exhibit only minor differences, leading to negligible reward variation and thus impairing effective decision-making.

To address these issues, we define the *cumulative Q-value trajectory score* as:

$$CQ_H(\tau) = \sum_{t=0}^{H} \gamma^t Q^\pi(s_t, a_t), \tag{24}$$

where $Q^\pi(s_t, a_t) = \mathbb{E}\left[\sum_{k=0}^{\infty} \gamma^k r_{t+k} \mid s_t, a_t\right]$ is the infinite-horizon action-value function.

### D.2    LEMMA AND PROOF

**Lemma D.1** (Weighting form of cumulative Q-value scoring). *Assume rewards are bounded, i.e., $|r_t| \le R_{\max}$. Then the expectation of the cumulative Q-value score satisfies:*

$$\mathbb{E}[CQ_H(\tau)] = \mathbb{E}\left[\sum_{j=0}^{H}(j+1)\gamma^j r_j + \sum_{j=H+1}^{\infty}(H+1)\gamma^j r_j\right]. \tag{25}$$

*Proof.* By definition,

$$\mathbb{E}[CQ_H(\tau)] = \mathbb{E}\left[\sum_{t=0}^{H} \gamma^t Q^\pi(s_t, a_t)\right] \tag{26}$$

$$= \mathbb{E}\left[\sum_{t=0}^{H} \gamma^t \mathbb{E}\left[\sum_{k=0}^{\infty} \gamma^k r_{t+k} \mid s_t, a_t\right]\right] \tag{27}$$

$$= \mathbb{E}\left[\sum_{t=0}^{H} \sum_{k=0}^{\infty} \gamma^{t+k} r_{t+k}\right]. \tag{28}$$

Since $\gamma \in (0, 1)$ and rewards are bounded, the infinite series is absolutely convergent, so we can safely exchange the order of summations:

$$\mathbb{E}[CQ_H(\tau)] = \mathbb{E}\left[\sum_{j=0}^{\infty} \left(\sum_{t=0}^{\min(H,j)} 1\right) \gamma^j r_j\right]. \tag{29}$$

The inner summation counts how many times each $r_j$ appears, which equals $\min(H, j) + 1$. Therefore:

$$\mathbb{E}[CQ_H(\tau)] = \mathbb{E}\left[\sum_{j=0}^{H}(j+1)\gamma^j r_j + \sum_{j=H+1}^{\infty}(H+1)\gamma^j r_j\right]. \tag{30}$$

$\square$

### D.3 INTERPRETATION

Equation equation 25 reveals the following properties:

- For $j \leq H$, each reward $r_j$ is upweighted by $(j+1)\gamma^j$, progressively emphasizing short-term rewards within the horizon.
- For $j > H$, each future reward is uniformly weighted by $(H+1)\gamma^j$, ensuring that long-term contributions are preserved rather than truncated.

Thus, the cumulative Q-value criterion serves as a structured approximation to the infinite-horizon RL objective:

$$R(\tau) = \sum_{t=0}^{\infty} \gamma^t r_t, \tag{31}$$

while mitigating short-horizon bias. Compared to the conventional finite-horizon score $\sum_{t=0}^{H} \gamma^t r_t$, it maintains **global sensitivity to future rewards**, better discriminates **high-potential trajectories**, and enhances both **prospective decision learning** and **safe exploration**.

## E TWO PROSPEC IMPLEMENTATION DETAILS

**Implementation Details of DMControl.** For continuous action spaces, we employ the SAC algorithm. The policy network in the SAC algorithm generates a probability distribution from which the actions are subsequently sampled. This policy network produces a probability density function that assigns probabilities to all potential actions based on a given state $s_t$. Using this distribution, we randomly draw $j$ actions to predict and execute prospective tasks.

$$\hat{a}_t^{0:j-1} = \{a_t^0, a_t^1, \cdots, a_t^{j-1}\} \sim \pi_\theta(|s_t) \tag{32}$$

**Implementation Details for Atari.** For discrete action spaces, we adopt the DQN framework as the policy method. In the DQN framework, the agent computes Q-values for all possible actions at the current state and selects the action with the highest Q-value. Thus, for ProSpec with discrete action spaces, we rank the Q-values of all actions in descending order and choose the top $j \in [1, n]$ actions to predict future states. This can be formalized as follows:

$$\hat{a}_t^{0:j-1} = \{a_t^0, a_t^1, \cdots, a_t^{j-1}\} = \arg \underset{a \in \mathcal{A}}{top} \, j \, Q(s_t, a) \tag{33}$$

The term "$arg \, top$" denotes the process of selecting the top $j$ actions that maximize the Q-values.

### E.1 HYPERPARAMETER SETTINGS

We list the hyperparameters used for the DMControl and Atari benchmarks in the Table 14 and Table 15, mainly following the CURL settings (Laskin et al., 2020a). The entire process is implemented using PyTorch, and network training is performed on an NVIDIA RTX 3090 GPU.

## F MORE EXPERIMENTAL RESULTS AND ANALYSIS

### F.1 MORE EXPERIMENTAL RESULTS

As shown in Table 5, we present all the results for the Atari benchmark. We also report the results for expert human and random play, as reproduced from (Yarats et al., 2020). Overall, ProSpec achieves an IQM HNS of 47.2%, outperforming various methods: 97.4% higher than SimPLe, 232.4% higher than DER, 153.7% higher than OTR and CURL, 92.6% higher than DrQ, 4.8% higher than SPR, 1.7% higher than VCR, 8.7% higher than PlayVirtual, 12.1% higher than PLASTIC, and 9.0% higher than RLTSC. In terms of individual games, ProSpec achieved the highest score in 7 out of 26 game scenarios. Notably, it demonstrated performance comparable to that of expert humans in certain environments, particularly in the *Bank Heist* scenario. In this scenario, ProSpec's score of 481.1 significantly outperforms other state-of-the-art models and approaches the expert human score of

753.1. The objective in *Bank Heist* is to evade the police and rob as many banks as possible, requiring the agent to predict the police's potential locations and actions to optimize the robbery route and avoid capture. Our method's emphasis on PT enables preemptive action planning, contributing to its superior performance over other models that lack this capability.

Table 5: Scores obtained by different methods on Atari with 100k interaction steps. ProSpec was run with 10 random seeds, and the overall IQM HNS scores were recorded for evaluation comparison.

| Game | Huam | Random | SimPLe | DER | OTR | CURL | DrQ | SPR | VCR | PlayVirtual | RLASTIC | RLTSC | ProSpec |
|---|---|---|---|---|---|---|---|---|---|---|---|---|---|
| Alien | 7,127.7 | 227.8 | 616.9 | 739.9 | 824.5 | 558.2 | 102.8 | 801.5 | 822.4 | **947.8** | 1032 | 1,129.8 | 862.8 |
| Amidar | 1,719.5 | 5.8 | 88 | 188.6 | 82.8 | 142.1 | 102.8 | 176.3 | 170.6 | 165.3 | 201.6 | 169 | **170.6** |
| Assault | 742 | 222.4 | 527.2 | 431.2 | 351.9 | 600.6 | 452.4 | 571 | 571.6 | 702.3 | **888.5** | 473.6 | 703.6 |
| Asterix | 8,503.3 | 210 | **1,128.3** | 470.8 | 628.5 | 734.5 | 603.5 | 977.8 | 1,071.5 | 933.3 | 1,066 | 734 | 957.4 |
| Bank Heist | 753.1 | 14.2 | 34.2 | 51 | 182.1 | 131.6 | 168.9 | 380.9 | 303.7 | 245.9 | 161.2 | 67.6 | **481.1** |
| Battle Zone | 37,187.5 | 2360 | 5,184.4 | 10,124.6 | 4,060.6 | 14,870 | 12,954 | 16,651 | 13,261 | 13,260 | 2,099 | 11,700 | **15,934.4** |
| Boxing | 12.1 | 0.1 | 9.1 | 0.2 | 2.5 | 1.2 | 6 | 35.8 | 42.5 | 38.3 | **44.5** | 11.9 | 41.5 |
| Breakout | 30.5 | 1.7 | 16.4 | 1.9 | 9.8 | 4.9 | 16.1 | 17.1 | 18.4 | 20.6 | **21** | 5.7 | 20.4 |
| Chopper Command | 7,387.8 | 811 | 1,246.9 | 861.8 | 1,033.3 | 1,058.5 | 780.3 | 974.8 | 1,024.2 | 974.8 | 891.2 | 1,120.6 | **1,351.0** |
| Crazy Climber | 35,829.4 | 10,780.5 | **62,583.6** | 16,185.3 | 21,327.8 | 12,146.5 | 20,516.5 | 42,923.6 | 40,048.4 | 23,176.7 | 31,223.8 | 12,810.1 | 38,803.0 |
| Demon Attack | 1,971 | 152.1 | 208.1 | 508 | 711.8 | 817.6 | 1,113.4 | 545.2 | 560.4 | 1,131.7 | **2,117.8** | 834.2 | 963.5 |
| Freeway | 29.6 | 0 | 20.3 | 27.9 | 25 | 26.7 | 9.8 | 24.4 | 18.7 | 16.1 | 27.1 | **28.1** | 25.1 |
| Frostbite | 4,334.7 | 65.2 | 254.7 | 866.8 | 231.6 | 1,181.3 | 331.1 | 1,821.5 | **2294.7** | 1,984.7 | 1,802.3 | 1,731.6 | 1,864.6 |
| Gopher | 2,412.5 | 257.6 | 771 | 349.5 | 778 | 669.3 | 636.3 | 715.2 | 539.7 | 684.3 | 839.4 | 682.4 | **866** |
| Hero | 30,826.4 | 1,027 | 2,656.6 | 6,857 | 6,458.8 | 6,279.3 | 3,736.3 | 7,019.2 | 5,838.8 | 7,007.2 | 7,394.4 | **8,597.5** | 8,140.5 |
| Jamesbond | 302.8 | 29 | 125.3 | 301.6 | 112.3 | **471** | 236 | 365.4 | 382.5 | 394.7 | 461.1 | 329 | 384.0 |
| Kangaroo | 3,035 | 52 | 323.1 | 779.3 | 605.4 | 872.5 | 940.6 | 3,276.4 | **3,393.1** | 2,384.7 | 1,636.1 | 1,402.8 | 2,494 |
| Krull | 2,665.5 | 1,598 | 4,539.9 | 2,851.5 | 3,277.9 | 4,229.6 | 4,018.1 | 3,688.9 | 4,199.2 | 3,880.7 | **5,019.5** | 3,665.7 | 4,542.2 |
| Kung Fu Master | 22,736.3 | 258.5 | 17,257.2 | 14,346.1 | 5,722.2 | 14,307.8 | 9,111 | 13,192.7 | 19,679.7 | 14,259 | 16,105 | 12,680.4 | **20,711** |
| Ms Pacman | 6,951.6 | 307.3 | **1,480** | 1,204.1 | 941.9 | 1,465.5 | 960.5 | 1,313.2 | 1,477 | 1,335.4 | 1,245.6 | 1,392.2 | 1,139.7 |
| Pong | 14.6 | -20.7 | **12.8** | -19.3 | 1.3 | -16.5 | -8.5 | -5.9 | 0.9 | -3 | -17.7 | -14.8 | -1.1 |
| Private Eye | 69,571.3 | 24.9 | 34.9 | 72.8 | 59.6 | 81.9 | 3.5 | 86 | 98.9 | 93.9 | **100** | **100** | **100** |
| Qbert | 13,455 | 163.9 | 1,288.8 | 1,152.9 | 509.3 | 1,042.4 | 854.4 | 669.1 | 791.1 | 3,620.1 | **3,986.3** | 2,170.8 | 3,950 |
| Road Runner | 7,845 | 11.5 | 5,640.6 | 9,600 | 2,696.7 | 5,661 | 8,895.1 | 14,220.5 | 10,746.1 | 13,534 | **15,073.8** | 15,040.5 | 13,744 |
| Seaquest | 42,054.7 | 68.4 | 683.3 | 354.1 | 286.9 | 384.5 | 301.2 | 583.1 | 521.2 | 527.7 | 635.9 | 396.8 | **800.4** |
| Up N Down | 11,693.2 | 533.4 | 3,350.3 | 2,877.4 | 2,847.6 | 2,955.2 | 3,180.8 | **28,138.5** | 14,674.1 | 10,225.2 | 66,473 | 4,072 | 16,640.3 |
| IQM HNS (%) | - | - | 23.9 | 14.2 | 18.6 | 18.9 | 24.5 | 45.0 | 46.4 | 43.4 | 42.1 | 43.3 | **47.2** |

## F.2 ANALYSIS OF PROSPEC'S APPLICABILITY

To underscore ProSpec's advantages, we conducted a rigorous head-to-head evaluation against the robust PlayVirtual baseline across several Atari titles. Both models were trained for 100,000 iterations under identical random seeds and hyperparameter configurations, with checkpoints preserved for downstream analysis. We then examined their behavior in controlled scenarios—namely, *Bank Heist, Chopper Command*, and *Amidar*—by initializing each environment to a fixed, predefined state. ProSpec and PlayVirtual were loaded separately to generate action selections, and for each agent we logged the chosen action alongside the resultant next state (see Figure 9). We note that these experiments were not performed in the DMControl suite, as the tasks in DMControl are less straightforward to interpret.

Across extensive experiments, we find that ProSpec consistently outperforms pure trial-and-error in three key domains:

**Deep Prospective Planning.** In *Bank Heist*, PlayVirtual greedily chases the immediate reward—moving directly toward the money—without accounting for the approaching guards, and is quickly captured. In contrast, ProSpec's FDM simulates multiple future trajectories and chooses to veer right around the guards, successfully avoiding capture. By generating several candidate paths and selecting the locally optimal, highest-reward action, ProSpec demonstrates far superior long-horizon planning compared to pure trial-and-error.

**High-Risk Trap Avoidance.** In *Chopper Command*—where a single mistake ends the episode—ProSpec looks several steps ahead to identify potentially lethal states. As shown in Figure 9b, when the helicopter nears ground targets, PlayVirtual blindly ascends and crashes, whereas ProSpec's lookahead evaluation triggers a timely shot and evasive turn, dramatically reducing resets and enhancing safety.

**Sparse-Reward, Goal-Driven Tasks.** In *Amidar*, rewards are exceptionally sparse and naive exploration converges slowly. Figure 9c illustrates how PlayVirtual, following a greedy single-step policy, blunders into densely packed enemies and is surrounded; ProSpec, however, leverages multi-step forecasting to veer right and avoid danger, sustaining survival longer and obtaining sparse rewards more quickly—thereby accelerating convergence to near-optimal performance.

By contrast, in dense-reward, reactive environments such as *Freeway*, pure trial-and-error already performs well, and ProSpec's extra lookahead yields only modest gains.

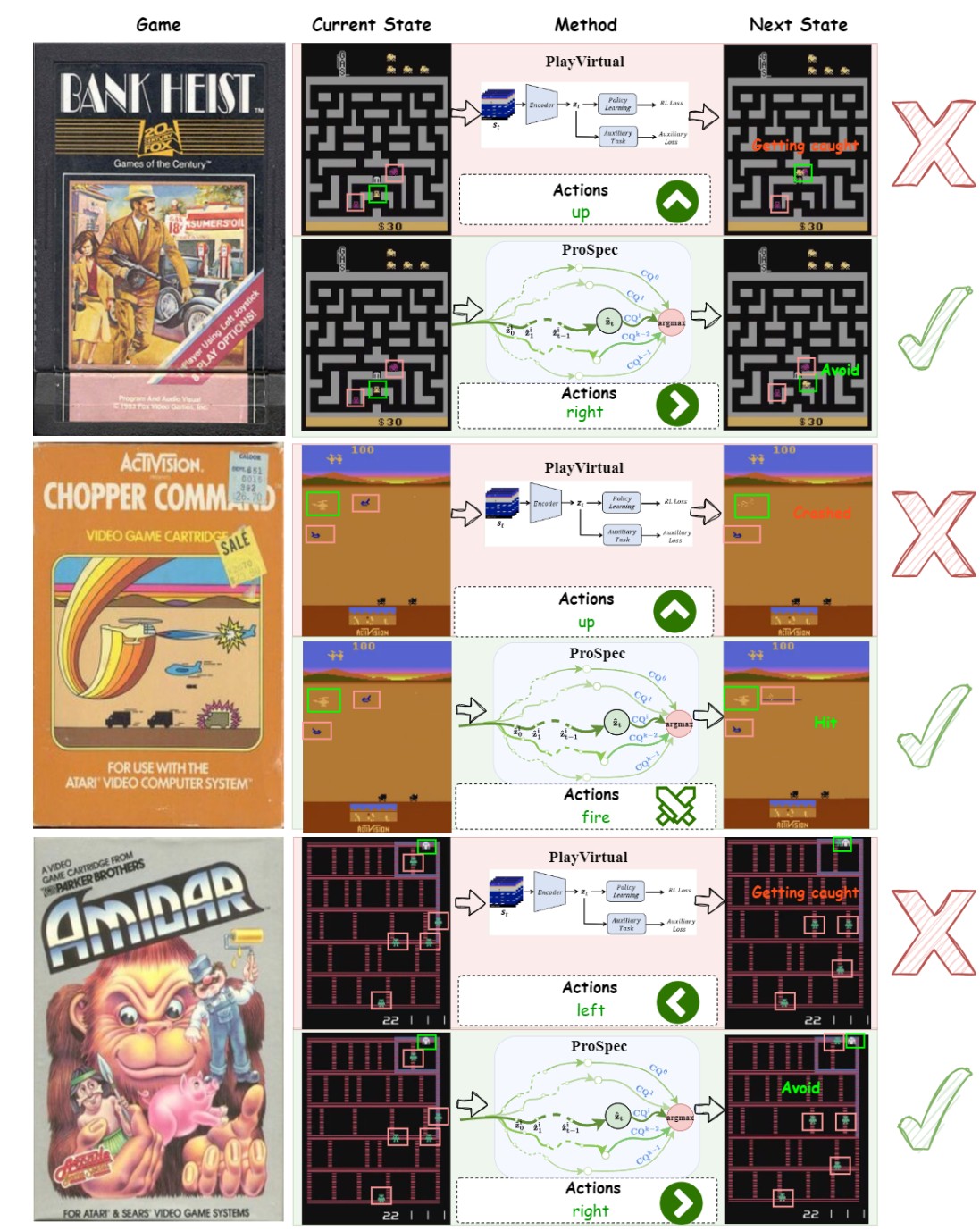

Figure 4: ProSpec vs. PlayVirtual Decision-Making Examples in Atari Games (Bank Heist, Chopper Command, Amidar).

### F.3 MORE ABLATION STUDIES

We conducted additional experiments to demonstrate the effectiveness of the proposed method. To account for the variance across different environments and mitigate the impact of computational limitations, we performed evaluations on the DMControl and Atari benchmarks using 5 random seeds.

**Effectiveness of Prospective Thinking.** Additional experiments were conducted to demonstrate the effectiveness of PT. Table 6 and Table 7 illustrate improvements in ProSpec across all environments. ProSpec-NP is the version without PT. PT resembles residual structures in neural networks from a behavioral standpoint, allowing the model to approach local optima with each decision. This suggests a consistent improvement in performance with each decision, akin to scenarios without PT. Hence, this capability significantly improves overall performance.

Table 6: Impact of PT on ProSpec in DMControl benchmark.

|  | ProSpec-NP | ProSpec |
|---|---|---|
| **Finger, spin** | 843±48 | 875±104 |
| **Cartpole, swingup** | 816±41 | 833±34 |
| **Reacher, easy** | 688±28 | 782±137 |
| **Cheetah, run** | 459±22 | 477±24 |
| **Walker, walk** | 455±56 | 507±60 |
| **Ball in cup, catch** | 884±68 | 927±17 |
| **Median Score** | 752 | **807.5** |

Table 7: Impact of PT on ProSpec in Atari benchmark.

| Game | ProSpec-NP | ProSpec |
|---|---|---|
| **Alien** | 794.9 | 862.8 |
| **Amidar** | 151.2 | 170.6 |
| **Assault** | 648.6 | 703.6 |
| **Asterix** | 958.3 | 957.4 |
| **Bank Heist** | 317.7 | 481.1 |
| **Battle Zone** | 12,334 | 15,934.4 |
| **Boxing** | 43.7 | 41.5 |
| **Breakout** | 18.8 | 20.4 |
| **Chopper Command** | 709 | 1,351 |
| **Crazy Climber** | 28,505.8 | 38,803 |
| **Demon Attack** | 898.9 | 963.5 |
| **Freeway** | 26.9 | 25.1 |
| **Frostbite** | 537.3 | 1864.6 |
| **Gopher** | 747.9 | 866 |
| **Hero** | 6,698.4 | 8,140.5 |
| **Jamesbond** | 351.5 | 384 |
| **Kangaroo** | 3,281 | 2,494 |
| **Krull** | 3,842.7 | 4,542.2 |
| **Kung Fu Master** | 17,756.2 | 20,711 |
| **Ms Pacman** | 1,202.7 | 1,139.7 |
| **Pong** | -4.6 | -1.1 |
| **Private Eye** | 100 | 100 |
| **Qbert** | 2,920.1 | 3,950 |
| **Road Runner** | 8,172.3 | 13744 |
| **Seaquest** | 607.3 | 800 |
| **Up N Down** | 8,977.3 | 16,640 |
| **IQM HNS (%)** | 43.5 | **47.2** |

**Impact of Prospective Count $k$.** We investigated the influence of the number of prospectives ($k$) on the performance of ProSpec. Experiments were conducted for $k \in [1, 9]$, recording median results across prediction horizons $t \in [1, 15]$. The findings, summarized in Table 8, include: (i) When $k = 3$, performance is lower compared to other $k$ values, highlighting the benefits of PT; (ii) In DMControl, the optimal performance is achieved at $k = 3$, with a 6.9% improvement over $k = 1$ and a 1.9% improvement over $k = 9$; (iii) In Atari, the highest performance occurs at $k = 9$, showing an 11% improvement over $k = 1$ and $k = 6$; (iv) Performance improvements do not scale linearly with the number of prospectives. In DMControl, when $k \geq 3$, performance improvements plateau, with

minimal further gains. This saturation may arise because, as $k$ increases, the sampled actions yield similar long-term rewards, resulting in negligible performance differences.

Table 8: Impact of prospective count $k$.

| Benchmark | k=1 | k=3 | k=6 | k=9 |
|---|---|---|---|---|
| **DMControl** | 745.9 | **797.0** | 795.1 | 781.8 |
| **Atari** | 37.3 | 39.7 | 37.3 | **41.4** |

**Impact of Prediction horizon** $t$. We investigated the impact of the prediction horizon ($t$) on ProSpec's performance, as presented in Table 9. Similar to the previous experiment, we assessed median performance across various values of $k$. The results can be summarized as follows: (i) In DMControl, ProSpec achieves optimal performance at $t = 6$, with a 6.3% improvement over $t = 1$ and a 6.2% improvement over $t = 15$; (ii) In Atari, ProSpec achieves the best performance at $t = 3$, with a 4.3% improvement over $t = 1$ and a 34.9% improvement over $t = 15$; (iii) Notably, ProSpec's performance does not increase linearly with the prediction horizon $t$. Instead, once a certain threshold is exceeded, the performance improvement begins to plateau or even decline. This phenomenon may arise because, as the prediction horizon increases, the complexity of future states that the model must capture also grows, leading to greater prediction errors and reduced accuracy.

**Overall.** Tables 10 to 13 illustrate the effects of the prediction horizon $t$ and prospective count $k$ on ProSpec performance across various environments. All results are summarized in Tables 8 and 9 for reference. Upon analysis, we determined that the optimal values for the prediction horizon and prospective count in the DMControl benchmark are $k = 3$ and $t = 6$, respectively. In the Atari benchmark, the optimal values are $t = 3$ for the prediction horizon and $k = 9$ for the prospective count.

Table 9: Impact of prediction horizon $t$.

| Benchmark | t=1 | t=3 | t=6 | t=9 | t=12 | t=15 |
|---|---|---|---|---|---|---|
| **DMControl** | 754.4 | 726.6 | **809.0** | 731.4 | 771.3 | 774.9 |
| **Atari** | 42.4 | **44.1** | 38.1 | 36.3 | 36.4 | 34.2 |

Table 10: Impact of prospective count $k$ in Atari-100k. Note that since Gopher has only 8 operable actions, the result for $k = 9$ is recorded as the result for $k = 8$.

| Game | k=1 | k=3 | k=6 | k=9 |
|---|---|---|---|---|
| Alien | 868.4 | 764.1 | 792.9 | 849.3 |
| Amidar | 149.5 | 115.8 | 132.5 | 208.7 |
| Assault | 629.8 | 610.8 | 650.1 | 653.0 |
| Asterix | 960.5 | 897.8 | 853.8 | 890.4 |
| Bank Heist | 273.8 | 225.0 | 258.0 | 585.1 |
| Battle Zone | 13,350.0 | 11,810.9 | 13,818.3 | 16,213.3 |
| Boxing | 35.6 | 28.1 | 35.4 | 41.1 |
| Breakout | 20.3 | 14.9 | 15.6 | 14.5 |
| Chopper Command | 597.0 | 860.9 | 1002.5 | 1064.3 |
| Crazy Climber | 25,691.0 | 22,536.2 | 19,337.3 | 15,873.0 |
| Demon Attack | 922.1 | 779.7 | 692.4 | 833.1 |
| Freeway | 29.4 | 26.9 | 27.7 | 23.6 |
| Frostbite | 267.8 | 1,805.0 | 1,990.9 | 1,632.7 |
| Gopher | 646.4 | 643.7 | 698.8 | 684.8 |
| Hero | 6,094.8 | 7,499.4 | 9,843.3 | 8,960.0 |
| Jamesbond | 373.5 | 307.4 | 336.9 | 348.6 |
| Kangaroo | 4,364.0 | 4,131.3 | 1,634.0 | 4,775.0 |
| Krull | 3,830.4 | 3,579.0 | 3,645.4 | 4,320.2 |
| Kung Fu Master | 14,497.0 | 16,651.1 | 16,439.8 | 12,952.8 |
| Ms Pacman | 1,154.5 | 964.6 | 980.6 | 1,041.3 |
| Pong | -8.4 | 3.9 | -12.7 | -13.0 |
| Private Eye | 100.0 | 100.0 | 105.5 | 100.0 |
| Qbert | 2,155.3 | 1,975.6 | 1,714.1 | 3,519.9 |
| Road Runner | 6,343.0 | 10,000.8 | 10,941.9 | 7,727.5 |
| Seaquest | 640.8 | 582.5 | 664.8 | 693.6 |
| Up N Down | 5,118.9 | 6,345.5 | 9,454.4 | 10,418.4 |
| IQM HNS (%) | 37.3 | 39.7 | 37.3 | **41.4** |

## F.4 THE IMPACT OF PROSPEC ON DATA EFFICIENCY.

We investigated the impact of PT on model convergence speed using the DMControl and Atari benchmarks, comparing the convergence performance of ProSpec and ProSpec-NP on a per-episode basis. For each environment in both benchmarks, we recorded the average rewards on the test set and the average game scores per episode. As shown in Figures 4-7, ProSpec demonstrated a significant improvement in data efficiency during the early stages of training, achieving much higher scores than ProSpec-NP. In terms of convergence, ProSpec reached near-optimal performance in a shorter time compared to ProSpec-NP. For instance, in the Amidar environment, ProSpec surpassed ProSpec-NP in score after fewer than 10,000 training steps, whereas ProSpec-NP took considerably longer to reach similar performance. PT enables ProSpec to prioritize actions that are safer and more likely to yield higher long-term rewards, allowing the model to rapidly acquire key strategies and show smooth performance improvement. In contrast, ProSpec-NP, which lacks PT, relies more on passive decision-making. This leads to slower learning and greater score fluctuations, especially in more complex environments. In terms of final model performance, ProSpec outperforms ProSpec-NP in nearly all environments, highlighting that PT significantly enhances the trial-and-error learning process of model-free methods, resulting in improved model performance. Overall, the ProSpec method, which incorporates PT, demonstrates superior convergence speed and final performance compared to ProSpec-NP, making it especially suitable for scenarios requiring fast learning and stable performance. This approach, which mimics human cognitive processes, has been shown to be both theoretically and practically effective.

Table 11: Impact of prediction horizon $t$ on ProSpec in Atari-100k.

| Game | t=1 | t=3 | t=6 | t=9 | t=12 | t=15 |
|---|---|---|---|---|---|---|
| Alien | 931.2 | 831.4 | 791.9 | 753.5 | 727.1 | 681.0 |
| Amidar | 168.5 | 152.2 | 160.1 | 132.1 | 125.6 | 155.1 |
| Assault | 686.3 | 663.6 | 660.9 | 678.3 | 642.2 | 460.4 |
| Asterix | 911.5 | 856.6 | 853.5 | 819.1 | 816.2 | 881.3 |
| Bank Heist | 272.8 | 372.4 | 493.4 | 180.3 | 181.5 | 303.0 |
| Battle Zone | 10,808.0 | 15,218.3 | 13,303.3 | 13,377.5 | 12,910.0 | 9,203.3 |
| Boxing | 35.1 | 32.7 | 43.7 | 35.4 | 39.2 | 19.9 |
| Breakout | 17.4 | 22.2 | 13.5 | 13.5 | 12.9 | 10.0 |
| Chopper Command | 729.0 | 1161.7 | 912.0 | 795.4 | 932.3 | 863.3 |
| Crazy Climber | 23,838.6 | 20,409.4 | 20,094.8 | 15,210.6 | 19,538.8 | 14,857.7 |
| Demon Attack | 793.3 | 902.0 | 852.8 | 737.3 | 638.9 | 463.2 |
| Freeway | 28.8 | 27.4 | 25.2 | 27.0 | 27.1 | 23.1 |
| Frostbite | 1,446.1 | 1,937.2 | 1,498.8 | 1,354.1 | 1,532.2 | 1,002.1 |
| Gopher | 710.1 | 836.5 | 721.0 | 669.8 | 517.5 | 422.3 |
| Hero | 7,962.0 | 5,874.2 | 8,426.9 | 8,408.5 | 9,537.9 | 6,335.0 |
| Jamesbond | 363.4 | 370.7 | 338.7 | 309.2 | 315.8 | 280.3 |
| Kangaroo | 2,337.8 | 2,461.3 | 3,058.0 | 3,993.3 | 2,896.7 | 4,454.0 |
| Krull | 4,136.2 | 4,296.7 | 3,450.2 | 3,690.2 | 3,322.8 | 3,966.3 |
| Kung Fu Master | 16,073.1 | 18,646.5 | 13,194.3 | 17,548.7 | 20,366.7 | 11,413.0 |
| Ms Pacman | 1,134.1 | 1,192.6 | 921.5 | 869.1 | 804.8 | 932.4 |
| Pong | -5.4 | -5.8 | -10.3 | -8.7 | -8.2 | -3.7 |
| Private Eye | 100.0 | 100.0 | 100.0 | 100.0 | 111.0 | 100.0 |
| Qbert | 3,307.4 | 3,019.5 | 1,833.8 | 2,189.2 | 760.0 | 1,070.6 |
| Road Runner | 9,833.1 | 7,431.2 | 9,314.3 | 11,405.6 | 11,167.7 | 7,705.7 |
| Seaquest | 605.3 | 711.3 | 604.5 | 614.5 | 682.1 | 677.3 |
| Up N Down | 17,462.2 | 7,222.8 | 2,907.5 | 4,238.3 | 9,431.6 | 4,767.2 |
| IQM HNS (%) | 42.4 | **44.1** | 38.1 | 36.3 | 36.4 | 34.2 |

Table 12: Impact of prospective count $k$ in DMControl-100k.

| Game | k=1 | k=3 | k=6 | k=9 |
|---|---|---|---|---|
| Finger, spin | 880.0 | 867.9 | 871.2 | 811.7 |
| Cartpole, swingup | 816.0 | 831.5 | 820.0 | 800.0 |
| Reacher, easy | 675.8 | 762.6 | 770.3 | 763.6 |
| Cheetah, run | 349.8 | 388.2 | 384.5 | 419.7 |
| Walker, walk | 424.6 | 561.4 | 517.6 | 516.5 |
| Ball in cup, catch | 845.1 | 905.5 | 905.8 | 934.4 |
| Median Score | 745.9 | **797.0** | 795.1 | 781.8 |

Table 13: Impact of prediction horizon $t$ on ProSpec in DMControl-100k.

| Game | t=1 | t=3 | t=6 | t=9 | t=12 | t=15 |
|---|---|---|---|---|---|---|
| Finger, spin | 877.6 | 813.0 | 820.6 | 785.3 | 772.3 | 810.4 |
| Cartpole, swingup | 813.0 | 793.0 | 818.0 | 817.0 | 829.0 | 836.0 |
| Reacher, easy | 695.8 | 670.4 | 785.2 | 675.3 | 766.7 | 699.5 |
| Cheetah, run | 361.7 | 375.3 | 466.5 | 415.3 | 368.6 | 382.9 |
| Walker, walk | 522.8 | 573.7 | 581.4 | 555.4 | 522.4 | 539.0 |
| Ball in cup, catch | 868.7 | 908.1 | 927.2 | 926.5 | 921.8 | 918.0 |
| Median Score | 754.4 | 726.6 | **809.0** | 731.4 | 771.3 | 774.9 |

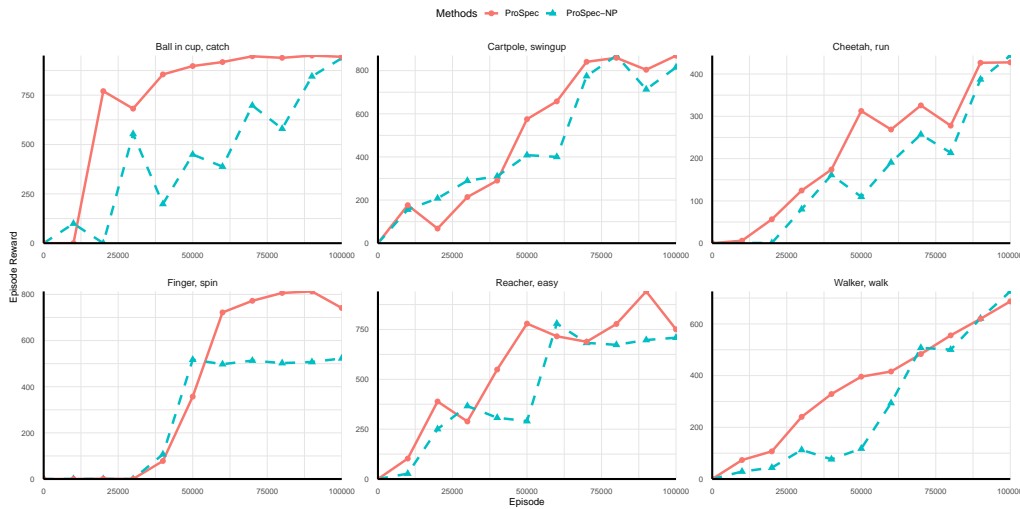

Figure 5: The impact of ProSpec on data efficiency (DMControl: Ball in cup to Walker).

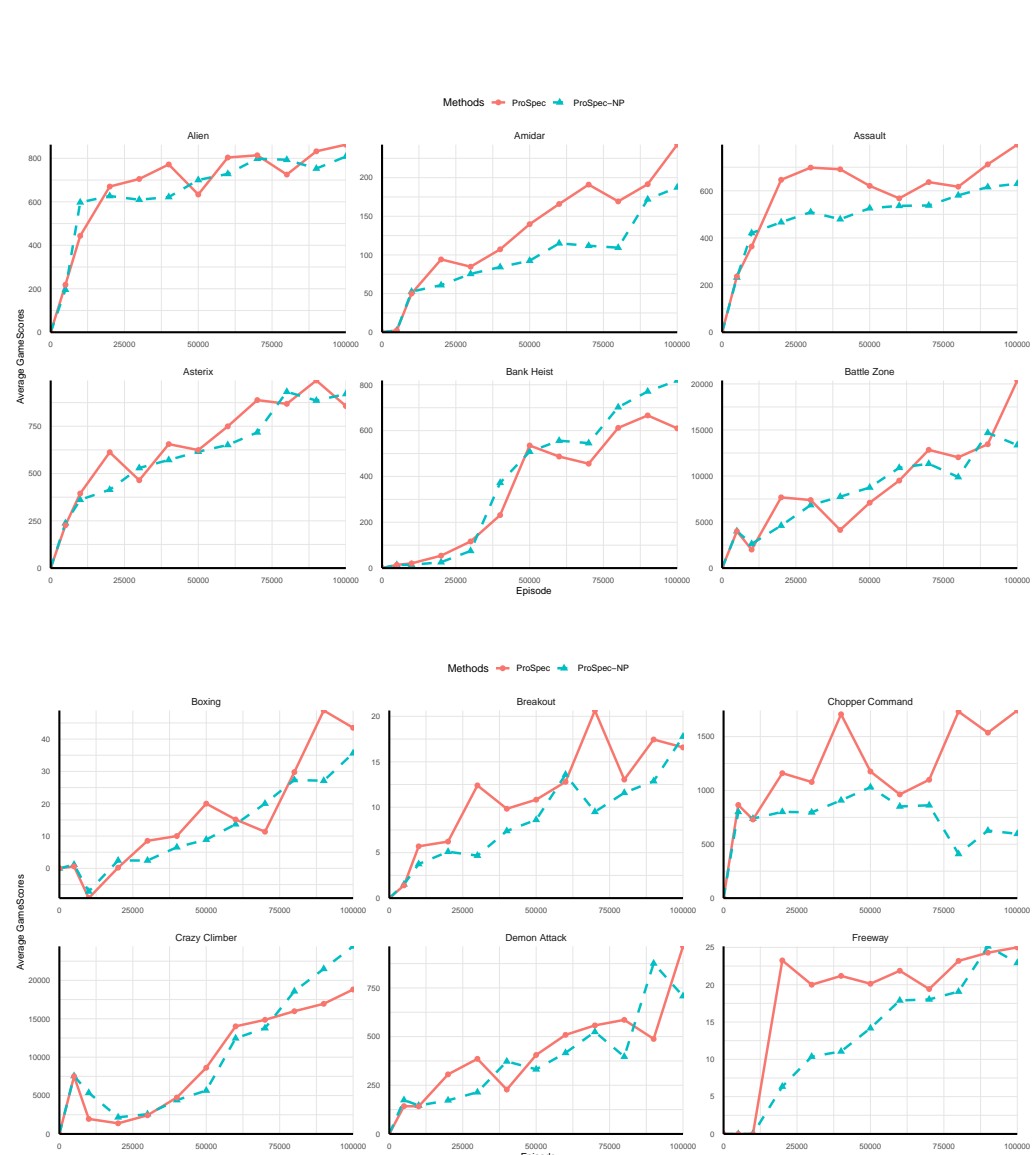

Figure 6: The impact of ProSpec on data efficiency (Atari: Alien to Freeway).

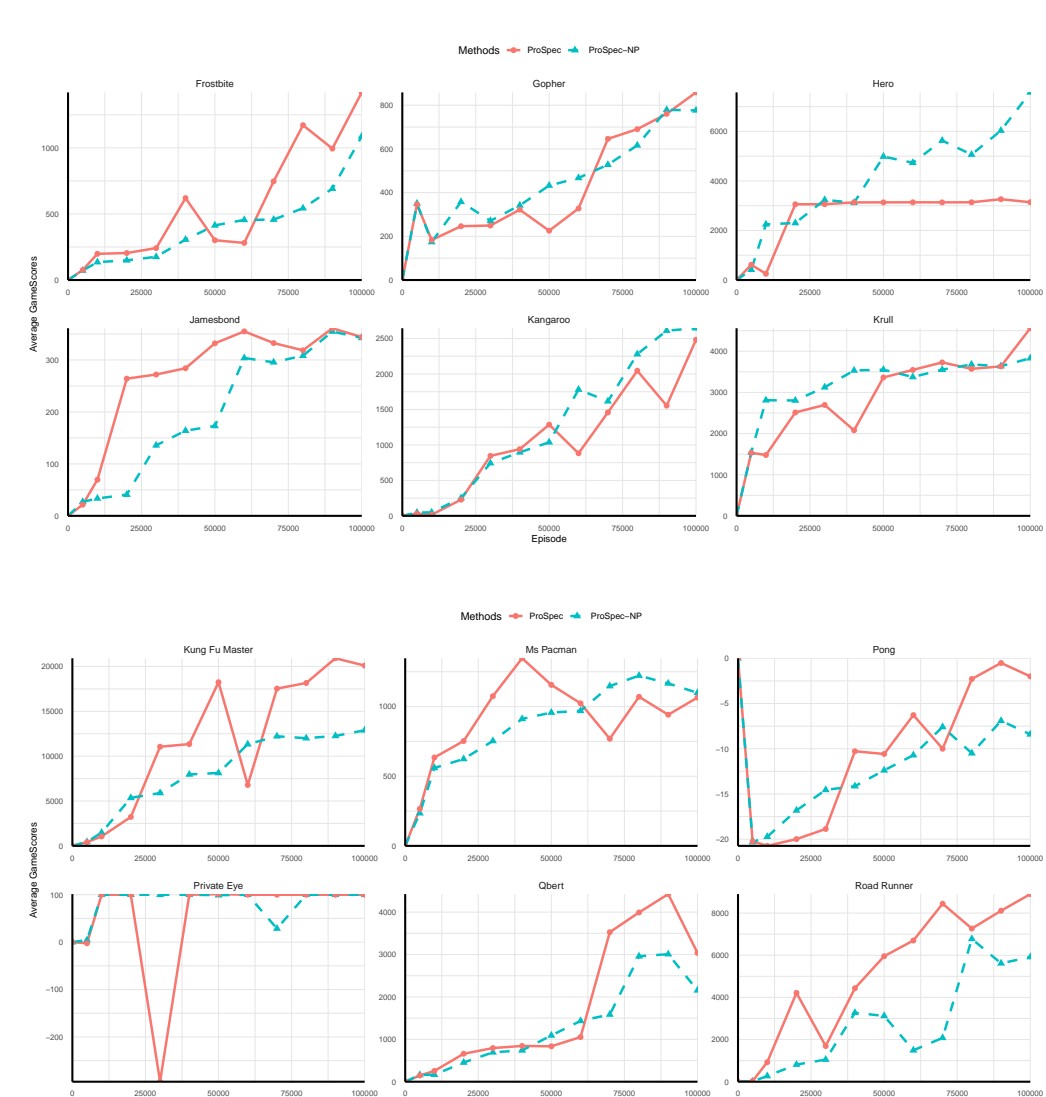

Figure 7: The impact of ProSpec on data efficiency (Atari: Frostbite to Road Runner).

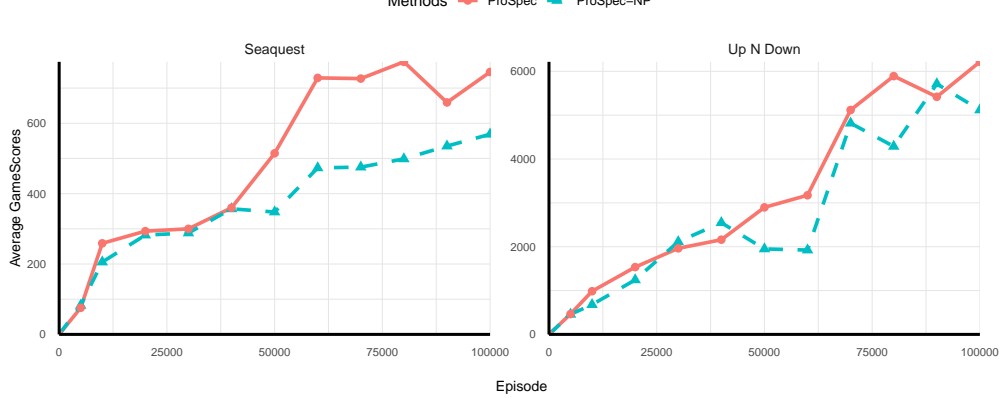

Figure 8: The impact of ProSpec on data efficiency (Atari: Seaquest and Up N Down).

### F.5 TIME COMPLEXITY

ProSpec augments a model-free learner with a FDM to enable multi-step prospective prediction. The additional computational overhead scales with the number of candidate trajectories $k$ and the prediction horizon $t$. By parallelizing the generation of the $k$ trajectories, we reduce the incremental cost to depend almost exclusively on $t$. To benchmark performance and time trade-offs, we ran all experiments in parallel on identical NVIDIA H100 GPUs using a fixed seed (1234). Figure 9 plots each method's evaluation-average reward (solid squares = ProSpec, dashed circles = PlayVirtual) against environment steps (bottom axis), with wall-clock training time to 100 K steps shown as vertical bars on the right axis.

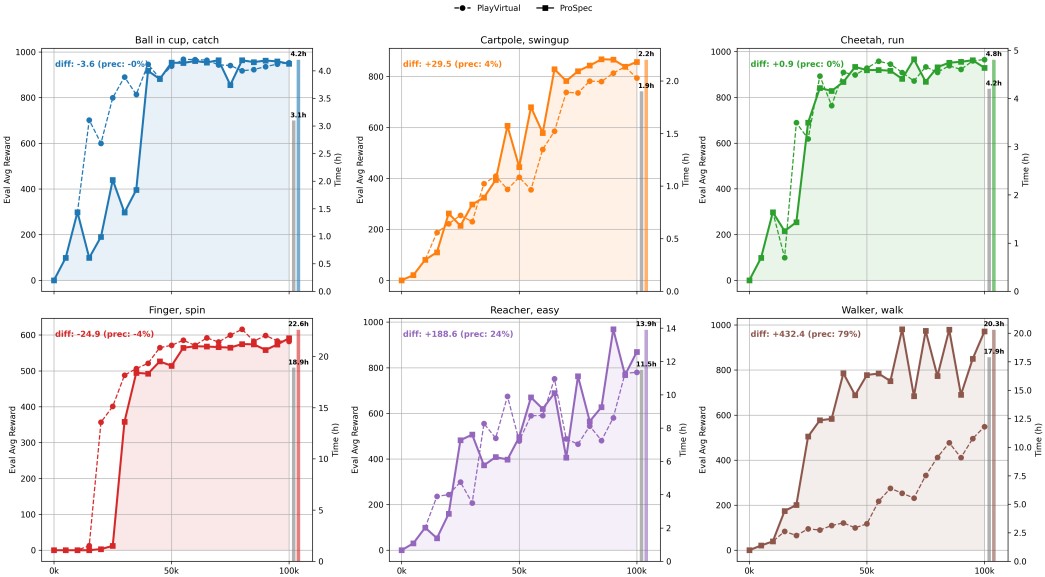

Figure 9: Time-performance comparison between ProSpec and PlayVirtual on six DMControl tasks. Curves show eval-average reward vs. environment steps; bars indicate cumulative training time to 100 K steps.

**Task-Dependent Gains vs. Overhead** Across six diverse DMControl benchmarks—from reflexive challenges to sparse-reward, high-dimensional tasks—ProSpec consistently unlocks significant performance improvements exactly where Prospective thinking is critical, at the cost of a modest $10 \sim 20\%$ increase in wall-clock training time:

- **Long-horizon locomotion** (*Walker, walk*): ProSpec secures the largest uplift—$+79\%$ ($+432.4$ pts)—for only a $14\%$ time penalty ($17.9\text{h} \rightarrow 20.3\text{h}$). This dramatic gain underscores its ability to anticipate complex gait dynamics over extended horizons.

- **Medium-horizon balancing** (*Cartpole, swingup*): A $15\%$ compute overhead ($1.9\text{h} \rightarrow 2.2\text{h}$) produces a $+4\%$ reward gain ($+29.5$ pts), validating ProSpec's Prospective thinking in refining multi-step control with minimal latency.

- **Sparse-reward, goal-driven tasks** (*Reacher, easy*): In an environment where rewards are exceedingly rare, ProSpec's multi-step Prospective thinking yields a $+24\%$ improvement ($+188.6$ pts) despite a $20\%$ time increase ($11.5\text{h} \rightarrow 13.9\text{h}$), markedly accelerating convergence to high-quality policies.

- **Reactive, dense-reward tasks** (*Ball in Cup, catch* and *Cheetah, run*): These tasks already converge rapidly under pure trial-and-error. ProSpec adds only 0.5–0.8 h of extra compute and matches peak performance within $\pm 1\%$, demonstrating negligible overhead when Prospective thinking is unnecessary.

- **High-dimensional dexterity** (*Finger, spin*): Despite a $20\%$ slower training curve ($18.9\text{h} \rightarrow 22.6\text{h}$), ProSpec achieves near-identical scores to PlayVirtual, highlighting a domain where future work on dynamics-model accuracy could unlock further planning benefits.

**Practical Trade-Off**   Although ProSpec can extend training time by up to 20% on H100 hardware, its double- and triple-digit reward gains in sparse-reward and long-horizon tasks make this overhead a highly favorable exchange. For real-world applications that demand Prospective thinking or robust risk avoidance, ProSpec's modest computational cost is readily justified by its substantial performance dividends.

**Extrapolating to Atari**   Although we have not directly profiled wall-clock overhead on the Atari benchmarks, the core extra cost of ProSpec—the multi-step FDM rollouts—depends only on the prediction horizon $t$ and is agnostic to whether the environment is continuous (DMControl) or discrete (Atari). In DMControl we observed a consistent 10–20% training-time overhead for horizons that yield the best performance gains. Since Atari agents operate at a similar per-step inference budget (frame-skip, network forward passes, etc.), we expect ProSpec to incur a comparable 10–20% overhead there as well. Crucially, in both domains that overhead is fully amortized by double- and triple-digit score improvements in sparse, long-horizon, or high-risk settings—suggesting that ProSpec's modest computational cost remains a sound trade-off across both continuous and discrete control benchmarks.

## G   POTENTIAL SOCIETAL IMPACT

Deep Reinforcement Learning (RL) has shown extensive potential across various domains like gaming, robotics, healthcare, and conversational systems. However, Model-free RL methods lack planning capabilities due to design differences from Model-based RL methods. To imbue Model-free methods with human-like PT abilities, we introduce ProSpec. This innovative approach makes optimal decisions by envisioning future n-stream trajectories, prioritizing higher value and lower risk outcomes. ProSpec also utilizes cycle consistency to address key RL challenges: augmenting state reversibility to avoid irreversible events (lower risk) and augmenting actions for improved data efficiency. Our validation in DMControl benchmark tests demonstrates superior performance. We anticipate ProSpec's impact on RL application development and hope it encourages further research into PT in RL. Responsible AI policies should guide image-based RL research and applications to ensure fairness and safety.

## H   THE USE OF LARGE LANGUAGE MODELS (LLMS)

Large Language Models (LLMs) were only used to assist in language polishing, such as grammar refinement and correction of typographical errors. They were not employed for idea generation, data analysis, or interpretation of results. All substantive contributions, including the conception, design, experiments, and writing of the manuscript, were carried out by the authors.

## I   REPRODUCIBILITY STATEMENT

We have made every effort to ensure the reproducibility of our results. Implementation details of the model, training procedures, and hyperparameter settings are provided in Appendix. All datasets used in our experiments are publicly available, with preprocessing steps consistent with PlayVirtual (Yu et al., 2021). To further facilitate reproduction, we provide our code and scripts as anonymous supplementary material at https://anonymous.4open.science/r/ProSpec-35B8/

Table 14: Hyperparameters for DMControl.

| Hyperarameter | Setting |
| --- | --- |
| Frame stack | 3 |
| Observation downsampling | (84, 84) |
| Augmentation | Random crop & intensity |
| Replay buffer size | 100K |
| Initial exploration steps | 1K |
| Action repeat | 2 finger, spin and walker, walk; 8 cartpole, swingup; 4 otherwise |
| Training steps | 100K |
| Evaluation trajectories | 10 |
| SAC Batch size | 512 |
| Q-function EMA $\tau$ | 0.01 |
| Critic target update freq | 2 |
| Discount factor | 0.99 |
| Initial temperature | 0.1 |
| Target network update period | 1 |
| Target network EMA $\tau$ | 0.05 |
| Actor & Critic & Encoder opt | |
| Optimizer | Adam |
| $(\beta_1, \beta_2)$ | (0.9, 0.999) |
| Learning rate | 0.0001 |
| Temperature $(\alpha)$ opt Optimizer | Adam |
| $(\beta_1, \beta_2)$ | (0.5, 0.999) |
| Learning rate | 0.0001 |
| cycle consistency batch size | 128 |
| $k$ (number of view angles) | 3 |
| $t$ (number of prediction horizon) | 6 |
| $M$ (number of virtual trajectories) | 10 |
| $\lambda_{pred}$ | 1.0 |
| $\lambda_c$ | 1.0 |
| warmup | Gaussian ramp-up $(i_{end} = 50K)$ |

Table 15: Hyperparameters for Atari.

| Hyperarameter | Setting |
|---|---|
| Gray-scaling | True |
| Frame stack | 4 |
| Observation downsampling | (84, 84) |
| Augmentation | Random crop & intensity |
| Action repeat | 4 |
| Training steps | 100K |
| Max frames per episode | 108K |
| Reply buffer size | 100K |
| Minimum replay size for sampling | 2000 |
| Mini-batch size | 32 |
| Optimizer | Adam |
| Optimizer:learning rate | 0.0001 |
| Opimizer:$\beta_1$ | 0.9 |
| Opimizer:$\beta_2$ | 0.999 |
| Opimizer:$\epsilon$ | 0.00015 |
| Max gradient norm | 10 |
| Update Distributional | Q |
| Dueling | True |
| Support of Q-distribution | 51 bins |
| Discount factor | 0.99 |
| Reward clipping Frame stack | [-1, 1] |
| Priority exponent | 0.5 |
| Priority correction | $0.4 \rightarrow 1$ |
| Exploration | Noisy nets |
| Noisy nets parameter | 0.5 |
| Evaluation trajectories | 100 |
| Replay period every | 1 |
| step Updates per step | 2 |
| Multi-step return length | 10 |
| Q network: channels | 32, 64, 64 |
| Q network: filter size | $8 \times 8, 4 \times 4, 3 \times 3$ |
| Q network: stride | 4, 2, 1 |
| Q network: hidden units | 256 |
| Target network update period | 1 |
| $\tau$ (EMA coefficient) | 0 |
| **Additional Hyperparameters in ProSpec** | |
| $M$ (number of virtual trajectories) | $2|\mathcal{A}|$ (two times of action space size) |
| $\lambda_{pred}$ (weight for prediction loss) | 1 |
| $\lambda_c^{max}$ (a weight related to cyclical consistency loss) | 1 |
| $k$ (number of view angles) | 9 |
| $t$ (number of prediction horizon) | 3 |
| $\lambda_{pred}$ | 1.0 |
| $\lambda_c$ | 1.0 |
| warmup | Gaussian ramp-up ($i_{end} = 50K$) |

