# OpenReview forum: "Prospective Decision Learning for Safe Exploration in Model-free Reinforcement Learning"
_ICLR.cc/2026/Conference — ICLR 2026 Conference Withdrawn Submission_

### Official Review · Reviewer_UFib · 2025-10-24

**Soundness:** 3
**Presentation:** 2
**Contribution:** 2
**Rating:** 2
**Confidence:** 4

**Summary:**

This paper introduces **ProSpec RL**, which applies prospective decision learning to model-free RL for safe exploration and state-trap escapement. In addition, the authors utilizes cyclical consistency regularization to improve data efficiency. The method is evaluated on DMControl and Atari100k benchmarks, showing improved performance compared to prior works.

**Strengths:**

**S1. Strong experimental results.**
* The method improves performance on both DMControl and Atari100k benchmarks.
* The presented empirical results support the claim that ProSpec RL enhances exploration efficiency and learning stability.

**S2. Clear organization and presentation.**
* The paper is clearly structured, which helps the reader follow the main ideas and technical contributions.

**Weaknesses:**

**W1. Concern about the framing of the work.**
* While the paper positions ProSpec RL as a model-free (or prospective) RL method, the approach relies on planning using learned world models.
* This design aligns more closely with model-based RL. The core novelty seems to lie in proposing a new planning objective (changing the return formulation from \sum \gamma^i (\sum_{j<i} r_j + Q_i}, to \sum \gamma^i Q_i.
* Moreover, notion of "safety" does not align with safe RL literatures.

* **I think the framing of the work should be reconsidered**, specifically clarifying 1) whether it is fundamentally model-free or model-based., 2) safety of the exploration.

**W2. Concern about the validity of the objective.**
* The new objective for planning also looks like a hack rather than a solid objective.
* Simply, it is removing the reward term from the $\lambda$-returns (with $\lambda=\gamma$), or weighting rewards differently as explained in Appendix D.3, which does not align with the true objective for RL.
* While authors claim that it helps the agent to escape from “state-trap”, and concrete example has been shown, I think we need a justification on this, specifically why this planning can still give the optimal policy (or near-optimal policy).

**Questions:**

**Q1. Clarification on the framing of the work**
* Given that the method involves planning with world models, should this work be categorized as model-based RL?
* Aligning with prior planning-based methods (e.g., TD-MPC[1] for planning, CoPlanner[2] for planning + exploration) could help contextualize the contribution more accurately.

**Q2. Theoretical justification**
* Is there a derivable lower bound or theoretical guarantee showing that optimizing the proposed objective approximates the optimal policy?

**Q3. Experimental baselines (Atari100k)**
* The Atari100k experiments include limited baselines.
* Including model-based works (e.g., DreamerV3) would provide a more comprehensive comparison, given the similarity in approach.

**Q4. Value-consistency constraint Could you clarify why this is referred to as a constraint?**
* It seems more like a modified objective term for planning rather than an explicit constraint.

**Q5. Definition of “safe” exploration I think the term safe exploration might be overstated.**
* The proposed mechanism appears to facilitate better exploration (via state-trap avoidance), but does not explicitly address safety as typically defined in safe RL literature.

---

### Official Review · Reviewer_KuXo · 2025-10-26

**Soundness:** 3
**Presentation:** 3
**Contribution:** 2
**Rating:** 2
**Confidence:** 2

**Summary:**

This paper proposes a decision-making method that looks several steps ahead to select the optimal action for interacting with the environment, thereby improving the agent’s decision quality. Meanwhile, the authors introduce a mechanism for both forward and backward state prediction to ensure that the future trajectories being considered are accurate and consistent with the environment dynamics.

**Strengths:**

1. The authors conduct extensive experiments.

2. The paper is relatively easy to understand.

**Weaknesses:**

1. My main concern with this paper is that although the authors claim their main contribution lies in introducing the idea of Prospective Thinking into reinforcement learning (RL), similar ideas have already been explored in several recent works over the past years—most of which refer to Prospective Thinking simply as Planning. That makes the paper limited in novelty.
2. The authors repeatedly state that previous works rely on trial-and-error, yet their own approach still essentially follows a trial-and-error framework.

**Questions:**

1. How does the Prospective Thinking proposed in this paper differ from the Planning used in previous works, and what are its advantages?

2. Compared with model-based methods, what advantages does this work offer in terms of data efficiency?

3. Fundamentally, isn’t this approach still a trial-and-error process?

---

### Official Review · Reviewer_sH8K · 2025-10-31

**Soundness:** 2
**Presentation:** 3
**Contribution:** 1
**Rating:** 2
**Confidence:** 3

**Summary:**

This paper introduces a Prospective Thinking (PT) mechanism into model-free reinforcement learning and proposes ProSpec. Specifically, the method simultaneously simulates K potential future trajectories, computes their discounted returns, and selects the initial action with the highest expected return. ProSpec incorporates a Feature Dynamics Model (FDM) with cycle-consistency constraints and Orthogonal Weight Normalization (OWN) to enhance representation learning.

**Strengths:**

1. The paper is well-organized and clearly structured. The writing is fluent and coherent, the motivation is clearly articulated, and the content is easy to follow.

2. The experiments are extensive and thorough, covering a wide range of settings. The results are analyzed in depth, with detailed and appropriate examination of the model architecture and hyperparameters.

3. The paper provides an anonymous code repository, pseudocode, and key hyperparameters, supporting good reproducibility.

**Weaknesses:**

1. The authors claim that this is the first introduction of a Prospective Thinking (PT) mechanism into a model-free RL method. However, LDCQ [1] has already incorporated the PT mechanism into a model-free RL framework through the sampling of target embeddings. \
[1] Venkatraman, Siddarth, et al. "Reasoning with Latent Diffusion in Offline Reinforcement Learning." The Twelfth International Conference on Learning Representations.

2. The novelty of this paper is limited to the online model-free RL setting. However, in broader RL paradigms, Prospective Thinking (PT) or mechanisms serving an equivalent role have been widely explored, such as in Diffuser[1], Decision Diffuser[2], ContraDiff[3], and LDCQ[4] mentioned in Weakness 1. The authors do not justify the necessity of specializing the PT mechanism to the model-free RL context.

[1] Janner, Michael, et al. "Planning with Diffusion for Flexible Behavior Synthesis." ICML, PMLR, 2022.\
[2] Ajay, Anurag, et al. "Is Conditional Generative Modeling All You Need for Decision Making?" ICLR 2023.\
[3] Shan, Yixiang, et al. "ContraDiff: Planning Towards High Return States via Contrastive Learning." ICLR 2025.\
[4] Venkatraman, Siddarth, et al. "Reasoning with Latent Diffusion in Offline Reinforcement Learning." ICLR 2024.\

3. The proposed method does not perform impressively, especially when compared with SOTA methods. For instance:

* In Finger, spin (100k steps), ProSpec outperforms the second-best method by only 9 points (0.9% of the maximum achievable score);
* In Ball in cup, catch (100k steps), by only 6 points (0.6%);
* In Finger, spin (500k steps), by only 9 points (0.1%).
Moreover, most of these improvements are within the variance range of the baselines and ProSpec itself. Besides, there is an error in Table 1: for Cheetah, run (100k steps), the best-performing method is Dreamer-V3 (536), which is much higher than ProSpec (477), yet ProSpec (477) is mistakenly marked as the best.

4. The paper does not discuss limitations and future work.

**Questions:**

1. How does this work differ from LDCQ [1]? \
[1] Venkatraman, Siddarth, et al. "Reasoning with Latent Diffusion in Offline Reinforcement Learning." The Twelfth International Conference on Learning Representations.

2. According to Section 4.3, FDM appears to serve primarily to enforce cycle consistency and consequently enhance representation learning. However, this introduces additional training losses and computational costs. Could the authors analyze these overheads and discuss whether the incurred costs are justified by the observed performance improvements?

3. In Section 4.3, the authors mention that “to ensure reversibility, we employ Orthogonal Weight Normalization (OWN).” Could the authors clarify what "reversibility " specifically means in this context, and explain why it is essential to the effectiveness of ProSpec?

4. Have the authors considered evaluating ProSpec on Mujoco tasks? I am curious about how ProSpec would perform on these classic control benchmarks.

---

### Note · Authors · 2025-11-12

I have read and agree with the venue's withdrawal policy on behalf of myself and my co-authors.